# Layer 2/3 pyramidal cells in the medial prefrontal cortex moderate stress induced depressive behaviors

Prerana Shrestha, Awni Mousa, Nathaniel Heintz*

Laboratory of Molecular Biology, Howard Hughes Medical Institute, The Rockefeller University, New York, United States

**Abstract** Major depressive disorder (MDD) is a prevalent illness that can be precipitated by acute or chronic stress. Studies of patients with Wolfram syndrome and carriers have identified *Wfs1* mutations as causative for MDD. The medial prefrontal cortex (mPFC) is known to be involved in depression and behavioral resilience, although the cell types and circuits in the mPFC that moderate depressive behaviors in response to stress have not been determined. Here, we report that deletion of *Wfs1* from layer 2/3 pyramidal cells impairs the ability of the mPFC to suppress stress-induced depressive behaviors, and results in hyperactivation of the hypothalamic–pituitary–adrenal axis and altered accumulation of important growth and neurotrophic factors. Our data identify superficial layer 2/3 pyramidal cells as critical for moderation of stress in the context of depressive behaviors and suggest that dysfunction in these cells may contribute to the clinical relationship between stress and depression.

*For correspondence: heintz@rockefeller.edu

Competing interests: The authors declare that no competing interests exist.

## Introduction

Major depressive disorder (MDD) is a prevalent and potentially life-threatening disorder that affects approximately 16% of the global population at some point in life (*Mayberg, 2009*; *Nestler and Hyman, 2010*; *Flint and Kendler, 2014*). Core symptoms of MDD include lack of motivation, reduced ability to derive pleasure from natural rewards (anhedonia) and abnormalities of sleep and appetite. MDD is thought to occur as a result of combined genetic, environmental, and biological factors. Although the nature of these factors and their relative contributions to the disorder remain largely unknown, human studies (*Mayberg, 2009*; *Hasler and Northoff, 2011*; *Morishita et al., 2014*) and circuit-based molecular genetic analysis in mice (*Nestler and Hyman, 2010*; *Li et al., 2012*; *Tye and Deisseroth, 2012*; *Svenningson et al., 2006*) have shown that MDD is characterized by alterations in the activity of a distributed circuit controlling emotive behaviors and that treatments that act within the medial prefrontal cortex (mPFC) can have a strong therapeutic effect (*Leuchter et al., 2012*; *Schmidt et al., 2012*). While these studies have demonstrated the complexity of the neural circuits controlling depressive behaviors and the actions of antidepressants, the impact of environmental factors on these circuits and their role in behavior has not been addressed adequately.

It is widely recognized that environmental stress, including early childhood trauma and recent stressful experiences, can contribute to MDD (*Russo et al., 2012*; *McEwen and Morrison, 2013*). Depressive behaviors can be elicited in experimental animals using a variety of stressors, which typically induce alterations in neuronal morphology and synaptic function (*Shansky and Morrison, 2009*; *Moench and Wellman, 2014*) in the mPFC that are consonant with the thinning of the mPFC that occurs in MDD (*Kroes et al., 2011*; *Grieve et al., 2013*). Inactivation of the ventral mPFC results in loss of cortical control of stress (*Diorio et al., 1993*; *Figueiredo et al., 2003*) and alters the activation of brainstem neurons that regulate behavioral responses to stress (*Maier, 2015*). Optogenetic activation of mPFC neurons projecting to the dorsal raphe nucleus can reversibly alter

**eLife digest** Around 16% of people will experience an episode of major depression at some point in their lives, with symptoms including a loss of motivation, a reduced enjoyment of previously pleasurable activities, and disturbances in sleep and appetite. Multiple genes and environmental factors have been implicated in depression, and one of the strongest risk factors for developing the disorder is exposure to stress.

Stress and depression affect many of the same brain regions, most notably the prefrontal cortex—an area that is involved in decision making, problem solving and regulating emotions. Shrestha et al. therefore reasoned that a good way of obtaining insights into the relationship between stress and depression would be to study prefrontal cortex cells that express genes that have been linked to depression.

One such gene is *Wfs1*. Mutations in this gene cause a rare disorder called Wolfram syndrome, in which affected individuals experience a wide range of symptoms that often include severe depression. Shrestha et al. identified a specific population of cells in the prefrontal cortex that express *Wfs1*. When subjected to a stressful event, such as being restrained, mice that had been genetically modified to lack this gene in their prefrontal cortex were more likely to exhibit depression-like behaviors than non-modified mice. The genetically modified mice also released more stress hormones when restrained and produced different amounts of a number of proteins that regulate the growth and signaling of neurons.

Shrestha et al. propose that these proteins act on neural circuits that control how the mice respond to stress. Furthermore, changes in the levels or the distribution of these proteins may increase the likelihood that a stressful event will trigger behaviors associated with depression. Further experiments are required to investigate the possibility that using drugs to manipulate cells that express *Wfs1* could protect against the harmful effects of stress, or even treat existing episodes of depression.

mobility in the forced swim test (FST), implicating this specific class of pyramidal cells in the behavioral responses to stressful situations (*Warden et al., 2012*). While these studies clearly establish the mPFC as a structure that is important in the generation and execution of depressive behaviors and stress responses, and in the therapeutic actions of deep brain stimulation and antidepressants such as selective serotonin reuptake inhibitors, they also highlight the anatomical and functional complexities of mPFC circuitry.

To investigate cortical cell types that may play additional roles in MDD, it would be informative to identify cell types in the prefrontal cortex that express specifically genes known to cause major depression and to assess their potential roles in the regulation of depressive behaviors. Although genome-wide association studies (GWAS) have demonstrated that common depression is likely to result from alterations in a very large number of genes of small effect (*Flint and Kendler, 2014*), studies of Wolfram syndrome have identified *Wfs1* as a clear example of a gene that can cause MDD in humans (*Swift et al., 1990*; *Crawford et al., 2002*; *Swift and Swift, 2005*). Here, we have employed bacTRAP translational profiling and virus mediated trans-synaptic tracing studies (*Wall et al., 2010*) to demonstrate that the Wolfram syndrome gene (*Wfs1*) is expressed in a specific population of neurotrophin 3 (*Ntf3*) and proenkephalin (*Penk1*) expressing pyramidal cells in superficial layer 2/3 of the mPFC. These cells receive projections from other cortical structures, from the posterior thalamic nuclear group, and from the lateral amygdala. Deletion of *Wfs1* in forebrain neurons of conditional knockout mice (Wfs1/CKO) alters stress-induced depression-related behaviors, induces the expression of the immediate early gene *Fos* in the paraventricular nucleus (PVN) of the hypothalamus and results in enhanced accumulation of serum corticosterone. Increased stress-induced depressive behavior is also evident in animals from which *Wfs1* was deleted specifically in the mPFC. *Wfs1* is present in the endoplasmic reticulum (ER) of superficial cortical pyramidal cells, and its loss in these neurons results in altered growth factor and neurotrophin processing in response to inescapable restraint stress. Taken together, our data demonstrate that superficial layer 2/3 pyramidal cells in the mPFC can play a critical role in stress-induced depressive behaviors and indicate that dysfunction of *Wfs1* in forebrain neurons leads to activation of the hypothalamic–pituitary–adrenal

(HPA) axis and elevated blood corticosterone levels. They highlight a role for superficial layer 2/3 pyramidal cells of the mPFC in the modulation of depressive behaviors and suggest that the major depression evident in patients with Wolfram syndrome and carriers may occur in part due to hyperactivation of the HPA axis in response to stress.

## Results

Wolfram syndrome is a complex, multisystem disorder that results in early onset diabetes, optic atrophy, and increased risk for MDD (*Rigoli et al., 2011*). Wfs1 knockout mice accurately model many aspects of Wolfram syndrome, including type 1 diabetes, retinal degeneration, and impaired behavioral responses to stress (*Luuk et al., 2008*; *Kato et al., 2008*). Given the dendritic and synaptic remodeling that occurs in layer 2/3 pyramidal cells in the mPFC in response to stress, and the role of the prefrontal cortex in the regulation of stress responsiveness, we were interested in comparative studies of *Wfs1* in frontal cortex pyramidal cell populations, and the impact of *Wfs1* deletion in these cell types.

### A bacTRAP transgenic line expressing in the supragranular layer of the prefrontal cortex

To target layer 2/3 pyramidal cells specifically in the mouse prefrontal cortex (*Figure 1A*), we took advantage of ISH data demonstrating that the gene for neurotrophin 3 (*Ntf3*) is expressed primarily in superficial layers of this region of cortex (*Figure 1B*) (www.bgem.com). bacTRAP transgenic lines expressing the EGFP/L10a fusion protein for use in translational profiling (*Doyle et al., 2008*; *Heiman et al., 2008*) were generated, and expression of the transgene assayed relative to the expression pattern of endogenous *Ntf3* as revealed by in situ hybridization. The Ntf3 bacTRAP founder line PS1046 was chosen for profiling studies because its expression in the mPFC (*Figure 1B–E*) and dentate gyrus (*Figure 1—figure supplement 1*) reproduced expression of the endogenous gene (*Figure 1B*). EGFP/ L10a expression was primarily restricted to superficial layer 2/3 in the mPFC, including the prelimbic, infralimbic and medial orbitofrontal regions. Expression of EGFP/L10a was observed only in the NeuN + neurons (data not shown). Furthermore, at high magnification, it was evident that the soma of EGFP/ L10a-expressing cells was pyramidal, with a clear apical dendrite (*Figure 1E*). EGFP/L10a expression was localized to the soma, proximal dendrites, and nucleolus, as described for other bacTRAP mouse lines (*Doyle et al., 2008*). These results confirm previous studies demonstrating laminar expression of *Ntf3* gene in the cortex (*Vigers et al., 2000*) and identify these cells as supragranular layer 2/3 pyramidal cells.

### Translational profiling of Ntf3 cortical cells

To determine whether *Wfs1* is expressed in the supragranular pyramidal neurons targeted in the Ntf3 PS1046 bacTRAP line and to gain insight into specifically expressed genes that might be altered in *Wfs1*-mutant cortical pyramidal cells, we employed TRAP translational profiling (*Heiman et al., 2008*) to compare these cells to other cortical cell types (*Doyle et al., 2008*). We chose for this analysis three previously characterized pyramidal cell types: GLT25D2 (layer 5 corticopontine projecting neurons expressing *Glt25d2*), S100A10 (layer 5 corticostriatal pyramidal cells expressing *S100a10*), NTSR1 (layer 6 corticothalamic pyramidal cells expressing *Ntsr1*); two broadly defined interneuron classes including DLX1 (non-fast spiking cortical inter-neurons targeted by *Dlx1*), and NEK7 (fast spiking cortical inter-neurons expressing *Nek7*); and two glial cell types, ALDH1L1 (astrocytes expressing *Aldh1l1*) and OLIG2 (oligodendrocytes characterized by expression of *Olig 2*) (*Figure 1F*). As expected from the anatomical characterization of the Ntf3-targeted cells, comparative analysis of TRAP data collected from these cell types (*Shrestha et al., 2015*; *Doyle et al., 2008*; *Schmidt et al., 2012*; *Nakajima et al., 2014*) indicates clearly that Ntf3 layer 2/3 pyramidal cells express genes in their ground state that can be used to distinguish them from other cortical cell types (*Figure 1G*).

Examination of mRNAs enriched in Ntf3 pyramidal cells (*Figure 1G,H*; *Figure 1—source data 1, 2*) relative to those expressed in the entire cerebral cortex reveals a variety of control genes previously shown to be expressed in layer 2/3 pyramidal cells, including *Cux2, Penk, Smoc2, Matn2, Wfs1, Htr5b, Nptx2, Hap1, Gsg1l, Ntf3* (www.brain-map.org). Laminar markers that have been reported to be expressed specifically in cortical layers 4 (e.g., *Hsd11b1*), 5 (e.g., *Etv1*), and 6 (e.g., *Drd1a, Ctgf*) were not enriched (data not shown). Of 9 annotated marker genes (*Aqp4, Gfap, Aldh1L1, Slc1a3, Olig2, Gad1, Dlx1, Nek7* and *Gad2*) for non-pyramidal cells (glial cells and inter-neurons), all were depleted in

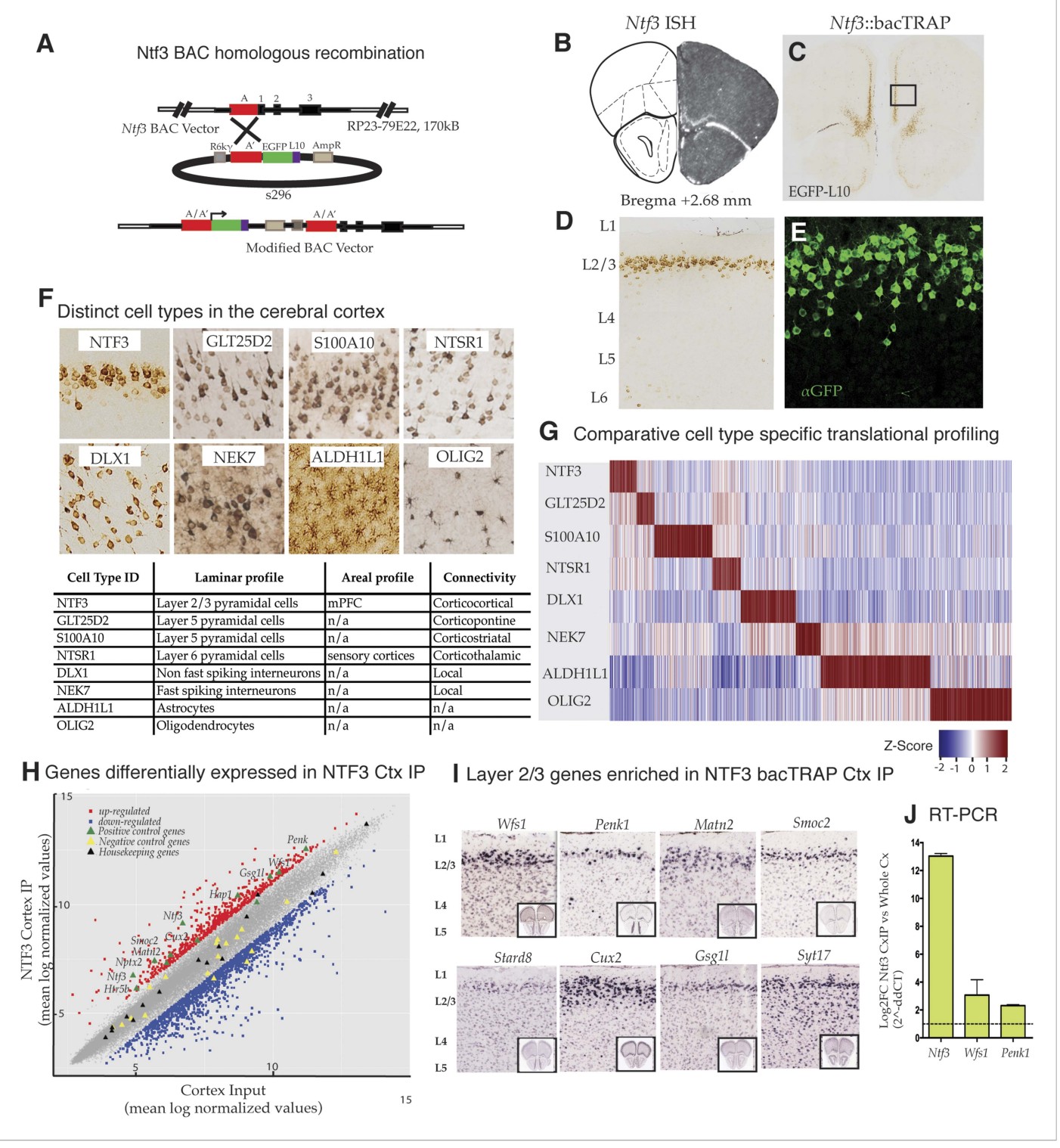

**Figure 1.** Identification and translational profile of a novel layer 2/3 pyramidal cell type in the cerebral cortex. (**A**) Ntf3 BAC RP23-79E22 was modified with an EGFP/L10 cassette using BAC homologous recombination methodology as previously described (*Gong et al., 2003*). EGFP/L10 expression in Ntf3:: bacTRAP line PS1046 recapitulates the pattern of endogeneous Ntf3 gene expression (**B**) (BGEM, ISH) in layer 2 of the medial prefrontal cortex (**C**, **D**). The EGFP/L10-positive cells in mPFC have pyramidal cellular morphology with an apical dendrite (**E**). Translational profiles of eight distinct cell types (**F**) in the cerebral cortex including distinct pyramidal cell types (NTF3, GLT25D2, S100A10, NTSR1), interneuron cell types (DLX1, NEK7), and glial cell types (ALDH1L1, OLIG2) were compared and shown in a heat map of cell-specific genes at 0.01 specificity index statistic threshold (**G**). Average normalized expression values of cell-specific enriched genes were z-transformed and plotted. Dark red = highly enriched. (**H**) Scatter plot of mean log normalized

*Figure 1. continued on next page*

*Figure 1. Continued*

values of NTF3 Cortex IP vs cortex input. Red and blue closed squares represent differentially expressed genes at 0.05 FDR corrected p-values and twofold change in expression. Yellow triangles represent negative control genes that are markers for non-neuronal cell types. Black triangles represent housekeeping genes whose expression is expected to be constant between the NTF3 cortex IP and cortex input. (**I**) Several layer 2/3 genes were enriched in NTF3 cortex IP such as *Wfs1*, *Penk1*, *Matn2*, *Smoc2*, *Stard8*, *Cux2*, *Gsg1l*, and *Syt17* (www.brain-map.org). (**J**) Quantitative RT-PCR independently confirm that *Ntf3* (2-ddCT = 12.09 ± 1.37), *Wfs1* (2-ddCT = 3.07 ± 0.86), and *Penk1* (2-ddCT = 2.69 ± 0.31) are enriched in the Ntf3 cortex IP.

The following source data and figure supplement are available for figure 1:

**Source data 1**. Top genes with the highest fold change difference in average expression between the NTF3 cortex IP and cortex input.

**Source data 2**. Genes highly ranked in NTF3 cell type by specificity index analysis across 8 cell types—GLT25D2 (corticopontine Layer 5 pyramidal cell type), S100A10 (corticostriatal Layer 5 pyramidal cell type), NTSR1 (corticothalamic Layer 6 pyramidal cell type), DLX (fast spiking inter-neurons), NEK7 (non-fast spiking inter-neurons), ALDH1L1 (astrocytes), and OLIG2 (oligodendrocytes).

**Figure supplement 1**. CNS expression pattern of EGFP.L10 in Ntf3 bacTRAP line PS1046.

the Ntf3 cells (*Figure 1H*). This is consistent with the anatomical studies of the Ntf3 line indicating that the targeted layer 2/3 cells are pyramidal. Housekeeping genes (*Rps6, Eif4e, Dnm2* and *Ttf1*) were all expressed within twofold enrichment boundary (*Figure 1H*). Furthermore, in situ hybridization data for Ntf3-enriched candidates from the microarray data reveal laminar expression (*Figure 1I*; http://mouse. brain-map.org/), and quantitative RT-PCR (qRT-PCR) of TRAP RNA provided additional confirmation that *Ntf3* (2-ddCT = 12.09 ± 1.37), *Wfs1* (2-ddCT = 3.07 ± 0.86), and *Penk* (2-ddCT = 2.69 ± 0.31) mRNAs are enriched in the Ntf3 cells (*Figure 1J*).

## Wolframin is present in the ER of layer 2/3 pyramidal cells

The demonstration that expression of *Wfs1* occurs in superficial layer 2/3 pyramidal cells in the prefrontal cortex of mice is interesting with regard to relationship between stress and depression. Recent studies have demonstrated convincingly that stress can induce dendritic remodeling of the prefrontal cortex (*Radley et al., 2004*) and that behavioral state–dependent synaptic modifications are important factors in the susceptibility and resilience to stress (*Wang et al., 2014*). Although the precise cell populations experiencing these changes in PFC have not been determined, the expression of *Wfs1* in layer 2/3 pyramidal cells and the clinical features of Wolfram Syndrome suggested to us that these cells may be of particular importance to the relationship between stress and depression. As a first step toward investigation of this possibility, we determined the subcellular distribution of Wolframin (WFS1), the protein encoded by the *Wfs1* gene, in the PFC. Thus, a specific antiserum to WFS1 was prepared and used to characterize its distribution in the PFC. As expected, WFS1 is expressed specifically in several structures in the adult mouse brain that are relevant to stress and depression, including superficial layers of the cerebral cortex, the central extended area of the amygdala (data not shown), and pyramidal cells in the CA1 field of the hippocampus (*Figure 2A–C*). Although WFS1 is clearly co-expressed with *Ntf3* in layer 2/3 pyramidal cells of the PFC (*Figure 2A*), its expression is entirely distinct from *Ntf3* expression in the hippocampus (*Figure 2C*).

WFS1 is an ER membrane protein (*Takeda et al., 2001*) that is thought to be important in $Ca^{2+}$ homeostasis (*Yurimoto et al., 2009*), insulin secretion (*Ishihara et al., 2004*), and the unfolded protein response (*Fonseca et al., 2005*). To determine whether the ER localization of WFS1 is maintained in the cerebral cortex, DAB-based pre-embedding immunoelectron microscopy was employed. Consistent with ultrastructural studies of peripheral and cultured cell types, dense staining of the ER was observed in all cortical cells that were positive for WFS1, principally in the cell soma and primary dendrites (*Figure 2D*).

## Selective deletion of *Wfs1* in cortical excitatory neurons

Wolfram syndrome is a complex, multisystem disorder whose features must reflect aberrant cell function in a number of tissues, including the pancreas and central nervous system (CNS). Given the prominent expression of *Wfs1* in the cortex, and the extensive literature documenting a role for the PFC in depression and resilience to stress, we were particularly interested in phenotypes that occur

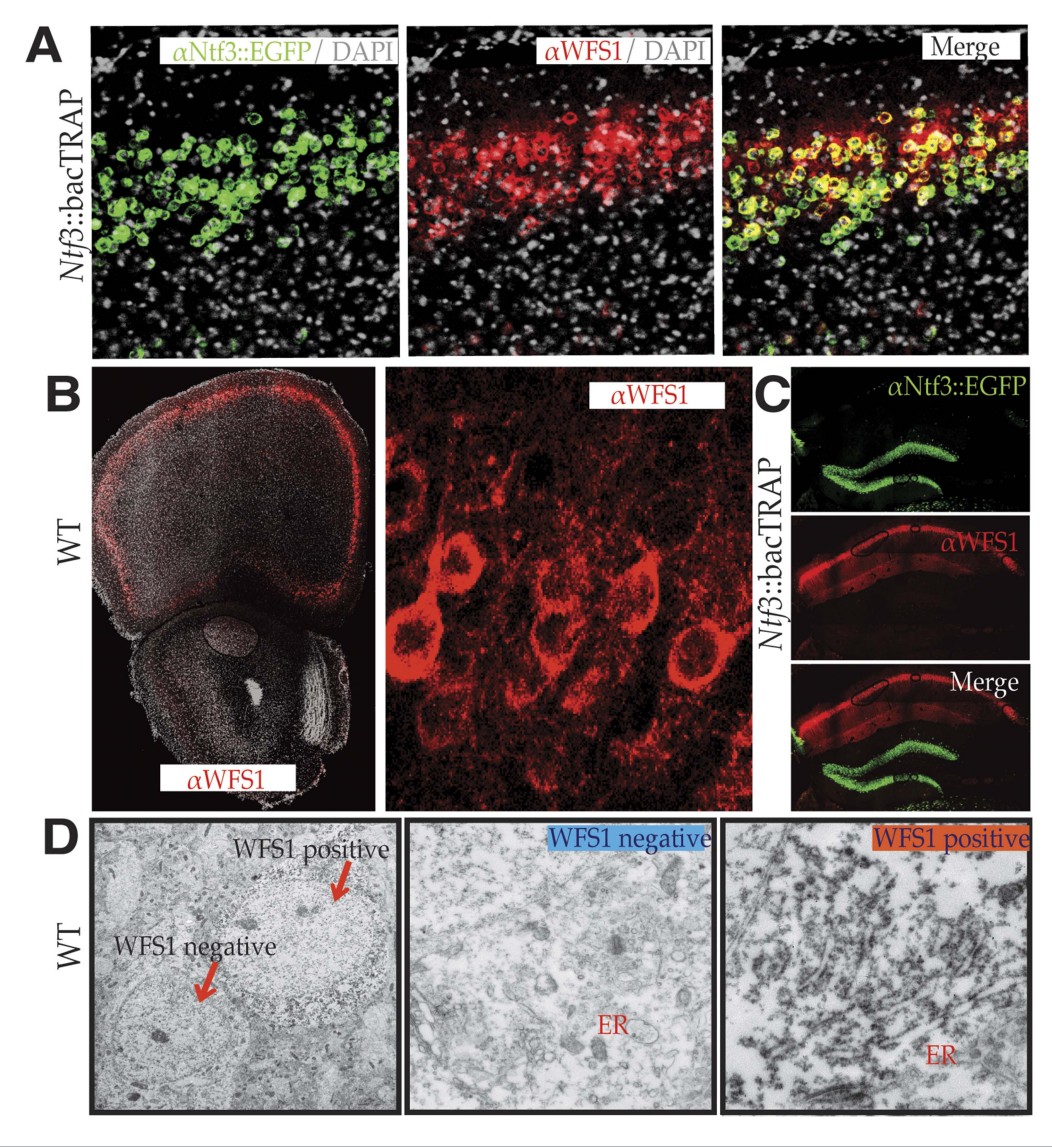

**Figure 2**. WFS1 is selectively enriched in NTF3 pyramidal cell population in the medial PFC. Using custom synthesized antiserum for WFS1, it is evident that the medial PFC EGFP/L10 neuronal population of Ntf3::bacTRAP transgenic mice co-immunostains for WFS1 (**A**). WFS1 is present in layer 2/3 of the cerebral cortex including medial PFC, and at higher magnification, it is evident that WFS1 is distributed in the cell soma as well as primary dendrites of layer 2 pyramidal cells (**B**). The co-expression of WFS1 and Ntf3::EGFP is region specific. In the hippocampus, WFS1 is expressed in CA1, whereas Ntf3::EGFP is expressed in the dentate gyrus (**C**). Immunoelectron micrography with WFS1 antiserum further shows that WFS1 is associated with rough endoplasmic reticulum in layer 2/3 pyramidal cells in the medial PFC neurons that express WFS1 (**D**).

specifically as a consequence of *Wfs1* mutation in the cortex. Accordingly, we generated a *Wfs1*-mutant mouse line carrying a conditional allele (Wfs1 F/F) by targeting loxP sites surrounding the eighth and largest exon of the gene (*Figure 3A*). This exon was chosen because it encodes the C-terminal hydrophilic domain, the site of many nonsense, and missense mutations that cause Wolfram Syndrome (*Rigoli et al., 2011*).

To remove WFS1 from the cerebral cortex, we employed the *Emx1*.IRES.cre driver line that expresses the Cre recombinase specifically in the telencephalon (*Figure 3B*) (*Gorski et al., 2002*). Important subcortical sites of Wfs1 expression, including the amygdala, striatum, and hypothalamus (*Luuk et al., 2008*), are unaffected in *Emx1*.IRES.cre driver line (*Figure 3B*; data not shown).

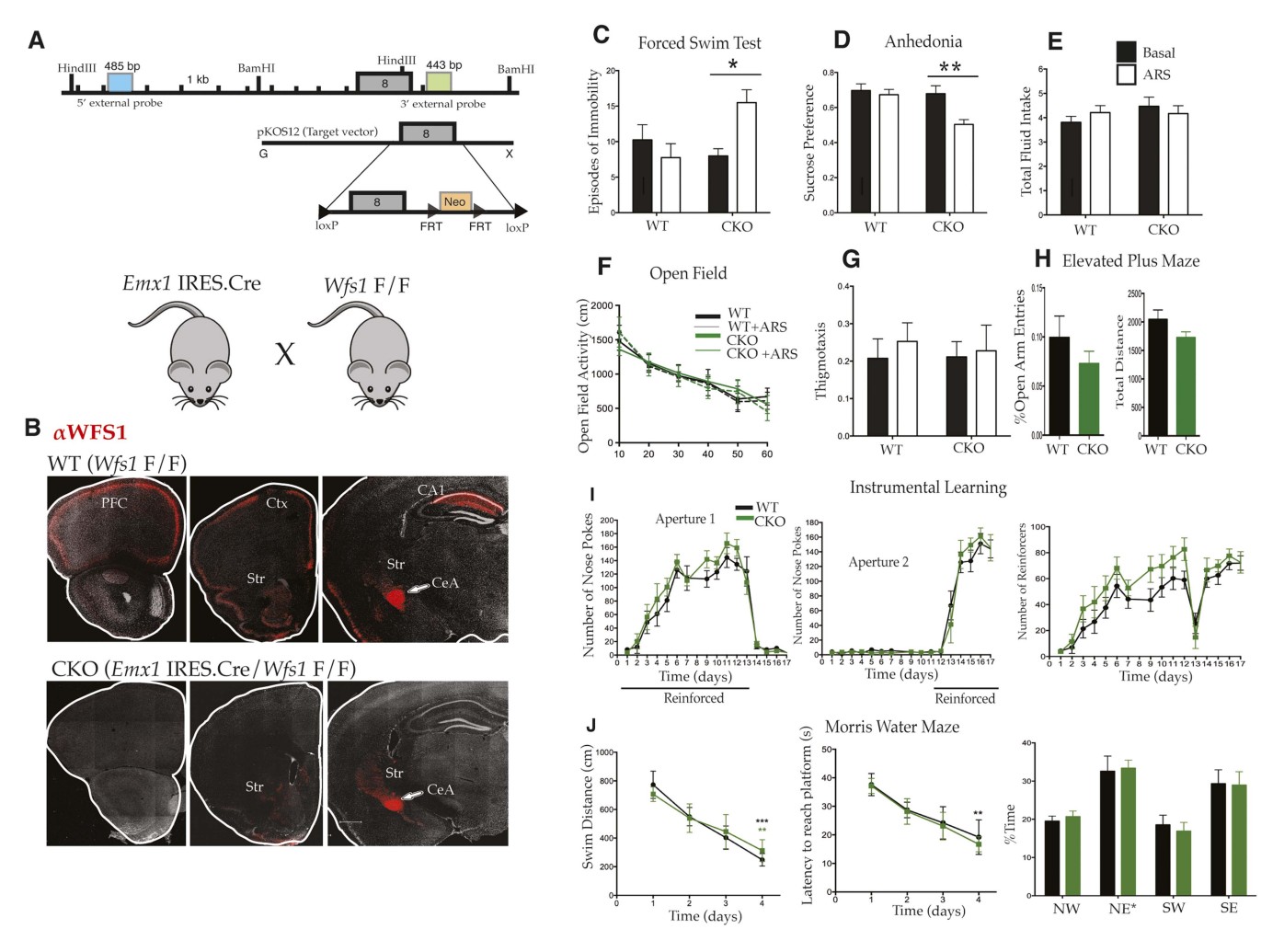

**Figure 3**. Selective deletion of Wfs1 in cortical excitatory neurons precipitates stress-induced depression-related behaviors. (**A**) Wfs1-mutant mice were generated by incorporating the conditional Wfs1-targeting vector that has the largest exon, exon 8, flanked by loxP sites. Wfs1 conditional KO (Wfs1/CKO) mice were produced by breeding Wfs1-mutant mouse bearing the Wfs1 conditional allele with Emx1.IRES.cre knock in mouse. (**B**) Normal pattern of WFS1 expression in the cerebral cortex (PFC, Ctx) and subcortical structures including striatum (Str), central amygdala (CeA), and hippocampus (CA1) is evident in wt (Wfs1 F/F) mice (top panel) whereas the cortical expression (cerebral cortex and hippocampus) of WFS1 is lost in Emx1.IRES.Cre Wfs1 F/F (Wfs1/CKO) mice. Wfs1 expression in CeA of Wfs1/CKO mouse brain is intact (bottom panel). (**C**) Wfs1/CKO mice exhibit equivalent episodes of immobility as WT mice in Porsolt's Forced Swim Test (FST) under baseline conditions but following acute restraint stress (ARS), they exhibit behavioral despair as indicated by increased episodes of immobility. FST: Two-way ANOVA—Bonferroni post hoc test: CKO Basal vs CKO + ARS, p < 0.05*, n = 8 per group. (**D**) Wfs1/CKO mice become anhedonic following ARS and lose normal preference for sucrose, but their total fluid intake is unchanged compared with WT regardless of exposure to stress (**E**). Sucrose preference test: Two-way ANOVA—Bonferroni post hoc test: CKO Basal vs CKO + ARS, p < 0.001**, n = 10 per group. Wfs1/CKO mice exhibit normal levels of spontaneous locomotion and acclimation to a novel arena in the open field test under baseline conditions and following ARS (**F**). Wfs1/CKO mice do not exhibit anxiety-related behavior following ARS as assayed by thigmotaxis in the open field arena (**G**). Likewise, the percentage time spent in open arms of the elevated plus maze is comparable for both genotypes (**H**). Wfs1/CKO mice display normal instrumental learning and reversal in three-choice operant chamber (**I**), and they display normal spatial learning and memory in Morris water maze in measures of swim distance, latency to reach platform, and % time spent in the probe quadrant NE post training (**J**).

Furthermore, although previous studies indicate that *Emx1* is expressed in all forebrain excitatory neurons and glia, the fact that Wfs1 expression is restricted to layer 2/3 cortical pyramidal cells and CA1 hippocampal neurons in forebrain structures indicates that any phenotypes evident in *Wfs1* telencephalon-specific conditional knockout mice (Wfs1/CKO) generated by conditional deletion of Wolframin using the *Emx1*.IRES.cre driver line must result from loss of function in these excitatory neuron populations. Wfs1/CKO mice were born in normal Mendelian ratio and had normal ad libitum

body weight compared with their wild-type (wt) littermates. Glucose levels were normal in these mice as assessed by urinalysis in non-fasting conditions (data not shown), indicating that the pancreatic β cells responsible for metabolic phenotype of Wolfram syndrome were spared in the Wfs1/CKO mice. These results are consistent with previous studies showing that *Emx1* is exclusively expressed in the CNS (*Gorski et al., 2002*).

## Wfs1/CKO mice display stress-induced depression-related behaviors

To assess the regional specificity of Wolframin loss in the Wfs1/CKO mice, the WFS1-specific antiserum was used for immunofluorescence localization studies. A normal pattern of WFS1 expression was evident in the Wfs1 F/F line, including expression in the cerebral cortex, the hippocampus, the amygdala (*Figure 3B*, top panel), and other selected subcortical structures (not shown). As expected, Wfs1/CKO mice carrying both the floxed allele and the *Emx1*.IRES.cre driver exhibited wt expression of WFS1 in all subcortical structures, but lost all detectable WFS1 expression in the cerebral cortex and hippocampus (*Figure 3B*, bottom panel).

The specific loss of *Wfs1* in the forebrain of Wfs1/CKO mice presented an important opportunity to determine whether the enhanced sensitivity to stress and depression of patients with Wolfram syndrome (*Koido et al., 2005*; *Swift and Swift, 2005*), or the behavioral abnormalities of *Wfs1* KO mice (*Kato et al., 2008*; *Luuk et al., 2009*), might result from abnormal function of forebrain neurons. Accordingly, Wfs1/CKO mice were put through a battery of behavioral tests to measure spontaneous activity, cognitive function, and depressive behaviors under normal baseline conditions, and in animals that had first been exposed to acute restraint stress (ARS). Acute psychological stressors such as swim stress and ARS strongly alter rodent behavior in the FST and in the sucrose preference test (SPT) (*Duncan et al., 1993*; *Cullinan et al., 1995*; *Jankord and Herman, 2008*). These tests are designed to measure two hallmarks of depression, behavioral despair, and anhedonia, and the effect of ARS on these behaviors in rodents has been used to model the effects of acute stress on depression (*Kohda et al., 2007*).

As shown in *Figure 3*, deletion of *Wfs1* from forebrain neurons in Wfs1/CKO mice revealed no differences in behavior under normal baseline conditions. Measurements of depressive behaviors (*Figure 3C,D*), fluid intake (*Figure 3E*), locomotion (*Figure 3F*), anxiety (*Figure 3G,H*), and cognitive function (*Figure 3I,J*) were all within the normal range for wt mice. However, the very specific effect of ARS on depressive behaviors was strongly altered in the Wfs1/CKO mice. Thus, Wfs1/CKO mice displayed a significant increase in immobility in the FST (*Figure 3C*; Two-way analysis of variance [ANOVA]—Bonferroni post hoc test: CKO Basal vs CKO + ARS, p < 0.05\*, n = 8 per group) and a strong decrease in sucrose preference after exposure to ARS stress (*Figure 3D*; Two-way ANOVA—Bonferroni post hoc test: CKO Basal vs CKO + ARS, p < 0.001\*\*, n = 10 per group). It is interesting that these stress dependent behaviors were observed despite the normal behavior of Wfs1/CKO mice in these tests in the absence of stress, and despite the fact that no effect of the loss of *Wfs1* in the cortex and hippocampus could be detected in the instrumental (*Figure 3I*) or spatial learning assays (*Figure 3J*). Our results indicate, therefore, that loss of *Wfs1* in forebrain neurons does not disrupt baseline learning or cognitive flexibility but renders the animals vulnerable to stress-induced depression-related behaviors.

## Deletion of *Wfs1* in the mPFC is sufficient to cause stress-induced depression-related behavior

Although the behavior of Wfs1/CKO mice clearly demonstrates that the loss of WFS1 in layer 2/3 cortical pyramidal cells and hippocampal CA1 neurons can result in enhanced responses to stress, previous studies have shown that the mPFC is both altered in response to stress (*Radley et al., 2004*) and important for behavioral resilience in response to stress under a variety of conditions (*Warden et al., 2012*; *Maier, 2015*). Accordingly, we were next interested in assessing whether the effects of stress on depressive behaviors in Wfs1/CKO mice mapped to the mPFC. To determine whether these behaviors are altered specifically because of abnormalities in the mPFC, we employed the viral Cre–loxP approach to knockout Wfs1 expression conditionally in a spatially and temporally controlled manner. Specifically, we injected an AAV2 expressing Cre recombinase tagged at the N-terminus with EYFP (AAV2.CMV.HI.GFP-Cre.SV40) into the mPFC of homozygous mutant mice (Wfs1 F/F) and generated Wfs1 mPFC.KO mice (*Figure 4A*). Control littermates (wt) were injected with an AAV2 expressing only EGFP, AAV2.CMV.PI.EGFP.WPRE.bGH.

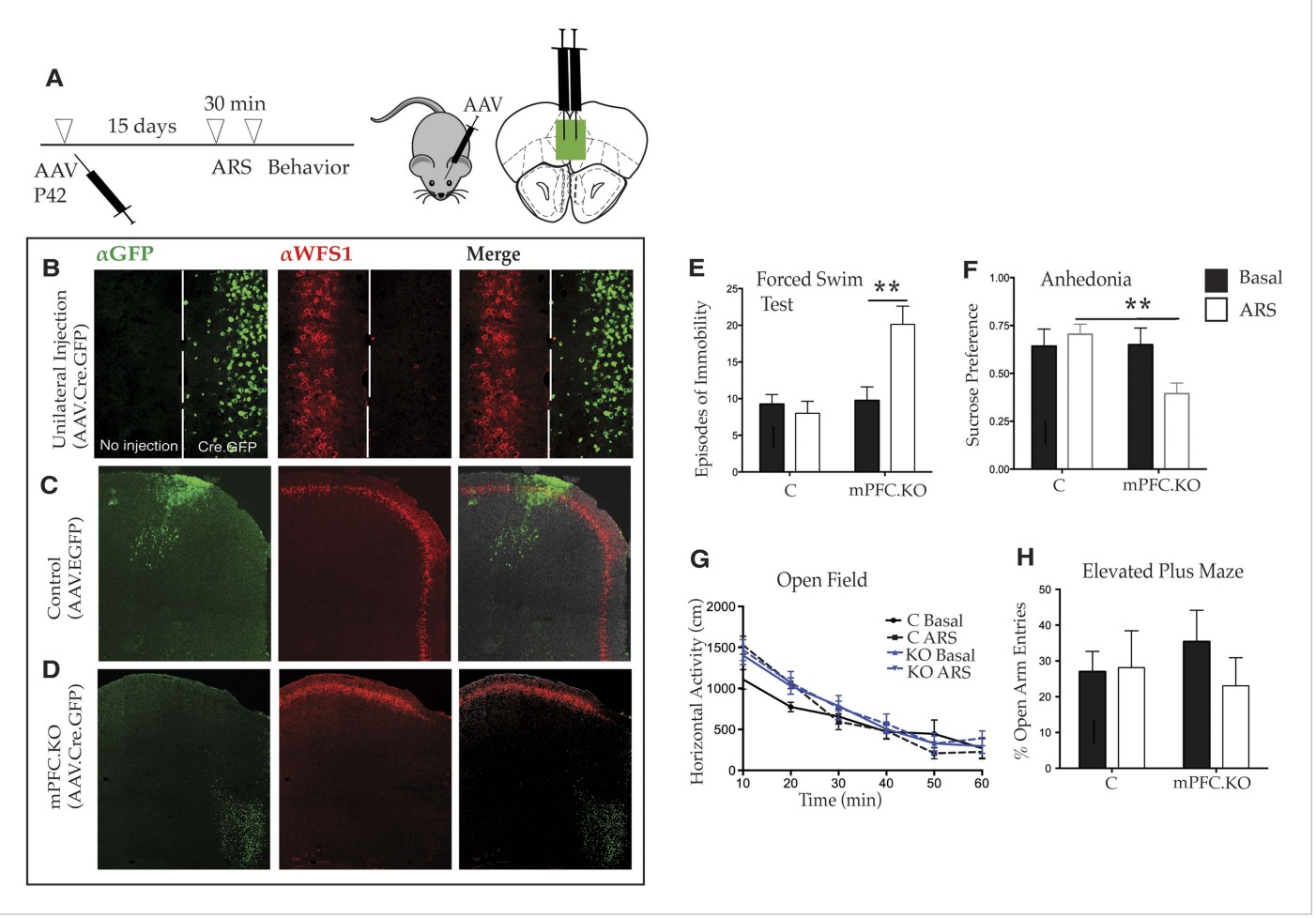

**Figure 4**. Deletion of Wfs1 in the medial PFC is sufficient to cause stress-induced depression-related behavior. (**A**) AAV2 expressing Cre recombinase tagged at the N-terminus with GFP (AAV2.CMV.HI.GFP-Cre.SV40) or control AAV2 expressing EGFP (AAV2.CMV.PI.EGFP.WPRE.bGH) was injected bilaterally in the medial PFC of Wfs1 F/F mice at P42 and allowed to express for 15 days. A subset of mice from both injected groups was subjected to 30 min ARS and 5 min short recovery followed by relevant behavioral assays. (**B**) Stereotaxic unilateral injection of AAV2.CMV.HI.GFP-Cre.SV40 in medial PFC demonstrates extremely high efficiency of the viral knockout strategy in deleting Wfs1 around site of injection. Compared with control vector injected mice which did not affect WFS expression, bilateral injection of AAV2.CMV.HI.GFP-Cre.SV40 results in loss of WFS1 in medial PFC (Wfs1/mPFC.KO) (**C** and **D**). (**E**) Under baseline conditions, Wfs1/mPFC.KO mice do not exhibit depression-related behaviors but following ARS, Wfs1/mPFC.KO mice exhibit a strong increase in episodes of immobility in Forced Swim Test (FST) (Two-way ANOVA—Bonferroni post hoc test: mPFC.KO Basal vs mPFC.KO + ARS, p < 0.01**, N = 8 per group). Wfs1/mPFC.KO mice also exhibit anhedonia following ARS in the sucrose preference test (SPT) (Two-way ANOVA—Bonferroni post hoc test: C + ARS vs mPFC.KO + ARS, p < 0.01**, N = 7 per group) (**F**). Wfs1/mPFC.KO mice have normal levels of spontaneous activity in the open field arena (**G**) and do not exhibit anxiety-related behavior in elevated plus maze as measured by percentage open arm entries (**H**).

Two sets of studies were conducted to demonstrate the local action of AAV2.CMV.HI.GFP-Cre.SV40 in vivo. First, the mPFC of Wfs1 F/F mice was unilaterally injected with the virus. As shown in *Figure 4B*, Wfs1 (red) continues to be expressed in the side of mPFC that has not received the virus, whereas Wfs1 expression is efficiently deleted on the side of the mPFC that was injected with AAV2.CMV.HI.GFP-Cre.SV40. Second, mice that were bilaterally injected with the Cre-expressing and control viruses were sacrificed and examined following their use for the behavioral studies. Although the use of intersectional viral approaches must necessarily result in some differences in the precise number and exact locations of the infected cells, examination of data from many animals indicated that there is no loss of Wfs1 expression in cortices injected with the control virus (*Figure 4C*), whereas loss of *Wfs1* in response to injection of the Cre-expressing virus was regional and reproducible (*Figure 4D*). Comparative analyses of these animals, therefore, can be used to determine whether Wfs1 is required specifically in the mPFC to moderate stress-induced depressive behaviors.

Accordingly, to investigate whether loss of *Wfs1* in the mPFC of Wfs1 mPFC.KO mice altered behavior in response to stress, a battery of behavioral tests was employed. As expected, there were no differences between control and Wfs1 mPFC.KO mice in any of the measured behaviors, including the behavioral despair and anhedonia assays, under normal baseline conditions (*Figure 4E*). However, exposure of Wfs1 mPFC.KO mice to ARS resulted specifically in enhanced responses to stress, including increased immobility in the FST (*Figure 4E*; Two-way ANOVA—Bonferroni post hoc test: mPFC.KO Basal vs mPFC.KO + ARS, p < 0.01**, N = 8 per group) and suppressed preference for sucrose (*Figure 4F*; Two-way ANOVA—Bonferroni post hoc test: C + ARS vs mPFC.KO + ARS: p < 0.01**, N = 7 per group). Spontaneous locomotion (*Figure 4G*) and anxiety-related behavior (*Figure 4H*) are not aberrant in the Wfs1 mPFC.KO mice after exposure to ARS. These data are consistent with those obtained from forebrain deletion of *Wfs1* in the Wfs1/CKO animals (*Figure 3C,D*) and extend those observations to allow the conclusion that Wfs1 function is required in the mPFC for layer 2/3 pyramidal cells to respond normally to stress. The observation that depressive behaviors are enhanced specifically in these mice strongly suggests that layer 2/3 pyramidal cells may be an important nexus for exploration of molecular mechanisms that contribute to the comorbidity of stress and depression in human populations.

## Wfs1/CKO mice display increased physiological responses to stress

Previous studies have demonstrated that both patients with Wolfram syndrome and *Wfs1* knockout mice display altered stress responses (*Sequeira et al., 2003*), including elevated levels of serum corticosterone upon exposure to stress (*Kato et al., 2008*; *Luuk et al., 2009*). Since *Wfs1* is expressed in many brain regions and in the periphery, we next sought to determine whether *Wfs1* expression in the forebrain contributes to the activation of the HPA axis and to the elevated corticosterone typical of animals that suffer from systemic loss of *Wfs1*. To explore this possibility, we examined the activation of HPA axis in wt and Wfs1/CKO mice after the application of ARS by cFOS immunostaining. As shown in *Figure 5*, ARS resulted in significantly higher levels of cFos activation in the hypothalamic PVN of Wfs1/CKO mice compared with controls (*Figure 5A,B*) (Two-way

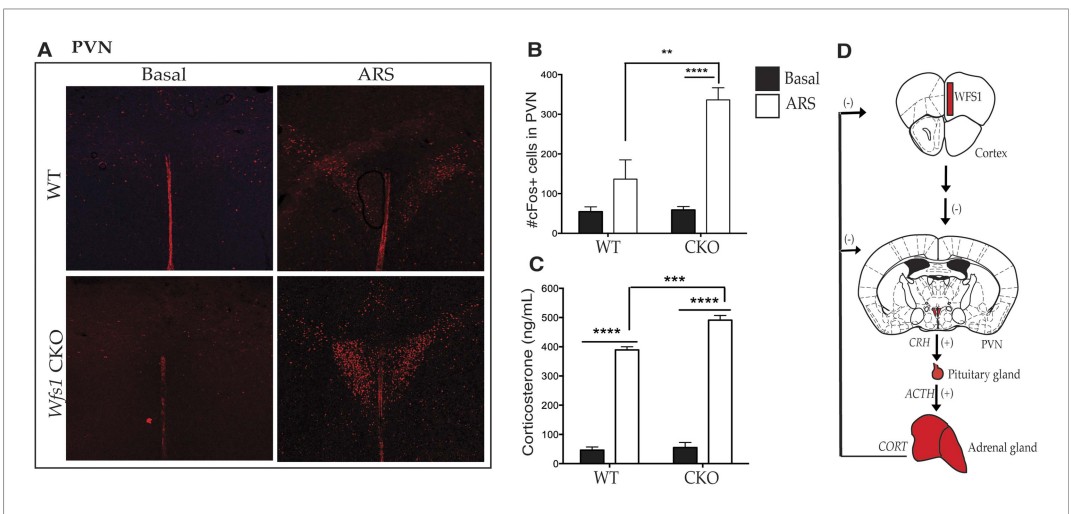

**Figure 5**. Wfs1/CKO mice display hyperactivation of the hypothalamic–pituitary–adrenal (HPA) axis in response to stress. Acute exposure to restraint stress significantly elevates cFos activation in the hypothalamic paraventricular nucleus in both groups of animals compared to basal conditions, but ARS also leads to a significantly higher increase of cFOS immunostaining in the PVN of Wfs1/CKO mice compared with controls (**A** and **B**) (Two-way ANOVA—Bonferroni post hoc test: WT + ARS vs CKO + ARS, p < 0.001***, N = 3 per group). Further downstream in the HPA axis, Wfs1/CKO mice release a significantly higher level of serum corticosterone (p < 0.001***) compared with controls (**C**) up on exposure to ARS (Two-way ANOVA—Bonferroni post hoc test: WT + ARS vs CKO + ARS, p < 0.001***, N = 5 per group). The intra-assay and inter-assay variability for the RIA were 5.08% CV and 1.99% CV, respectively. Our data support a model in which loss of Wfs1 in forebrain neurons causes hyperactivation of the PVN and the HPA axis is response to stress (**D**).

ANOVA—Bonferroni post hoc test: WT + ARS vs CKO + ARS, p < 0.001*** , N = 3 per group). Since the activation of the PVN is known to stimulate the pituitary gland to produce adrenocorticotropic hormone, which in turn induces corticosterone release from adrenal glands, we examined endocrine responses to stress in Wfs1/CKO mice using radioimmunoassay (RIA). We measured serum levels of corticosterone in Wfs1/CKO and control mice under basal conditions and after 30 min of ARS (*Figure 5C*). These experiments revealed that baseline corticosterone levels were not different in wt and Wfs1/CKO mice (WT: 46.17 ± 5.00; CKO: 55.26 ± 10.18) and that in both wt and Wfs1/CKO serum corticosterone levels increased in response to stress. However, the magnitude of the increase of serum corticosterone observed in Wfs1/CKO in response to stress (CKO + ARS: 505.12 ± 7.63) was significantly greater than that seen in their wt littermates (WT + ARS: 401.50 ± 6.00) (Two-way ANOVA—Bonferroni post hoc test: WT + ARS vs CKO + ARS, p < 0.001*** , N = 5 per group). The intra-assay and inter-assay variability for the RIA were 5.08% coefficient of variation (CV) and 1.99% CV, respectively. We conclude, therefore, that loss of *Wfs1* in forebrain neurons causes hyperactivation of the PVN and the HPA axis in response to stress (*Figure 5D*). Taken together with the enhanced behavioral responses of Wfs1/CKO and Wfs1 mPFC.KO mice to ARS, these results demonstrate that normal responses to stressful stimuli require the function of *Wfs1* in layer 2/3 pyramidal cells in the mPFC (*Figure 5D*). Given the identification of *Wfs1* as a genetic cause of MDD (*Flint and Kendler, 2014*) and previous studies demonstrating that behavioral resilience requires normal function of the mPFC (*Russo et al., 2012*), our data provide strong evidence that dysfunction in *Wfs1*-expressing pyramidal cells in the mPFC may contribute to MDD as a result of enhanced susceptibility to stress.

## Connectivity of *Wfs1*-expressing pyramidal cells in the mPFC

To gain additional insight into stress-induced depressive behaviors, it is important to identify additional features of the circuitry that underlie this behavior. As an initial effort to map specific circuit elements, we have used two approaches to discover brain areas and cell types that connect to *Wfs1*-expressing pyramidal cells in the mPFC (*Figure 6A*). First, we have employed a cre-dependent AAV reporter virus (AAV2.EF1a.DIO.eYFP.WPRE.hGH) to map projections originating in *Wfs1* pyramidal neurons expressing CreER$^{T2}$ (Wfs1::CreER$^{T2}$) in the mPFC. Injection ofAAV2.EF1a.DIO.eYFP.WPRE. hGH into the mPFC of Wfs1::CreER$^{T2}$ mice fed a tamoxifen diet results in recombination of the viral genome only in Wfs1 pyramidal cells, thus allowing identification of target sites receiving projections from these neurons. As shown in *Figure 6*, *Wfs1*-expressing neurons in the mPFC project locally within the mPFC, and across the corpus callosum to the contralateral primary motor cortex. These neurons also project subcortically through the pyramidal tract to terminate in the caudate putamen (*Figure 6B*). Second, we have used the two-virus monosynaptic circuit tracing approach (*Wall et al., 2010*) to identify cells that make direct, monosynaptic projections to Wfs1::CreER$^{T2}$ cells in the mPFC. A helper virus expressing Cre-dependent eGFP and an avian receptor protein, TVA (AAV9-pEF1a-FLEX-GT), was injected into mPFC of Wfs1:CreER$^{T2}$ mice, and allowed to express for 6 weeks, followed by the injection of glycoprotein deleted rabies virus expressing mCherry [(EnvA)SAD-dG-mCherry] that allows specific retrograde labeling of neurons making direct monosynaptic connections to the Wfs1 cells in mPFC. As shown in *Figure 6C*, our results demonstrate that Wfs1 pyramidal cells receive afferent information from pyramidal cells spanning several layers and regions of the cerebral cortex and that projection neurons in the posterior thalamic nuclear group (Po) and the lateral amygdala synapse onto Wfs1-expressing cells in the mPFC.

## *Wfs1* is required for normal ER function in the prefrontal cortex

One of the hallmark features of Wolfram syndrome is the presence of insulin-dependent diabetes. Studies of WFS1 localization and function in the pancreas have demonstrated that WFS1 is present in the ER, and thus it participates in a regulatory pathway that negatively regulates ER stress signaling (*Fonseca et al., 2010*; *Oslowski and Urano, 2011*). In the absence of *Wfs1*, pancreatic β cells are unable to adjust to changing glucose levels, resulting in altered β-cell function, apoptosis, and pancreatic degeneration (*Ishihara et al., 2004*). Given the present demonstration that WFS1 is also localized in the ER of layer 2/3 pyramidal cells in the mPFC (*Figure 2*) and that *Wfs1* deletion from these neurons leads to abnormal behavioral responses to stress and hyperactivation of the HPA axis (*Figures 3–5*), we were interested in assessing functions of PFC neurons that might be altered in response to loss of *Wfs1*. We chose to focus on neuropeptide and growth factor processing because of the role of WFS1 in insulin processing and secretion and because layer 2/3 pyramidal cells express a variety of peptides that have been implicated in stress and depression.

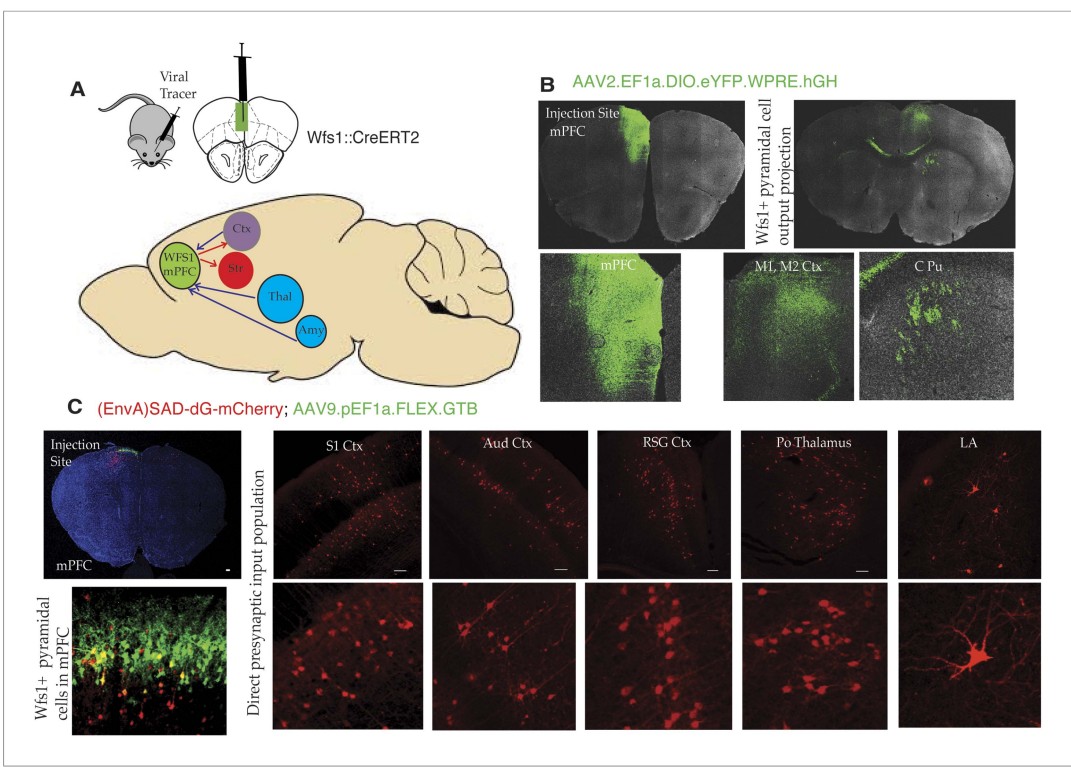

**Figure 6**. Connectivity of Wfs1-expressing pyramidal cells in the medial PFC. (**A**) Two different viral tracer strategies were applied to identify projections originating in mPFC Wfs1 pyramidal neurons (output) and monosynaptically connected presynaptic partners (input). (**B**) Cre-dependent AAV reporter virus (AAV2.EF1a.DIO. eYFP.WPRE.hGH) was injected in the mPFC of Wfs1::CreERT2 mice to map projections of Wfs1 pyramidal neurons expressing CreERT2 to the contralateral motor cortex (M1, M2) and caudate putamen (CPu). (**C**) Two-virus monosynaptic circuit tracing approach was used to identify direct presynaptic partners of Wfs1-expressing cells in the mPFC. A helper virus expressing Cre-dependent eGFP and TVA (AAV9-pEF1a-FLEX-GT) was injected into mPFC of Wfs1:CreERT2 mice and allowed to express for 6 weeks, followed by injection of glycoprotein deleted rabies virus expressing mCherry [(EnvA)SAD-dG-mCherry] that allows specific retrograde labeling of neurons making direct monosynaptic connections to the Wfs1 cells in mPFC. This strategy reveals that Wfs1 pyramidal neurons receive direct presynaptic input from neuronal populations in cortical areas including primary somatosensory cortex (S1 Ctx), auditory cortex (Aud Ctx), retrospenial granular cortex (RSG Ctx), posterior thalamic nuclear group (Po), and lateral amygdala (LA).

TRAP translational profiling of Ntf3 cortical cells (*Figure 1G*) demonstrated significant enrichment of the mRNAs for several secreted proteins that are of interest because of their known roles in neuronal function, including wingless-related MMTV integration site 7a (WNT7A) and neurotrophin 3 (NTF3). To determine whether their processing and/or abundance changed as a consequence of loss of *Wfs1* in the PFC, Western blots for each of these proteins were processed using brain extracts collected from wt and Wfs1/CKO mice under baseline conditions and following ARS. As a comparison, we assessed the abundance of brain-derived neurotrophic factor (BDNF) which is present in the PFC but not specifically enriched in the *Ntf3/Wfs1*-expressing cells. As shown in *Figure 7*, no differences were evident in any of these proteins from wt or Wfs1/CKO animals under baseline conditions. However, in response to ARS, the abundance of WNT7A and NTF3 were both altered. WNT7A is a secreted signaling factor that is involved in axonal remodeling and synaptic differentiation (*Ciani et al., 2011*). In Wfs1/CKO animals, it is specifically decreased in the PFC (Two-way ANOVA—Bonferroni post hoc test: CKO Basal vs CKO + ARS, $p < 0.001$** , N = 3 per group) (*Figure 7A*). Mature NTF3 (~14 kDa) is processed from a ~32-kDa precursor that is referred to as proNTF3 (*Yano et al., 2009*). Although we could not reliably detect mature NTF3 on Western blots from PFC, increased levels of proNTF3 were

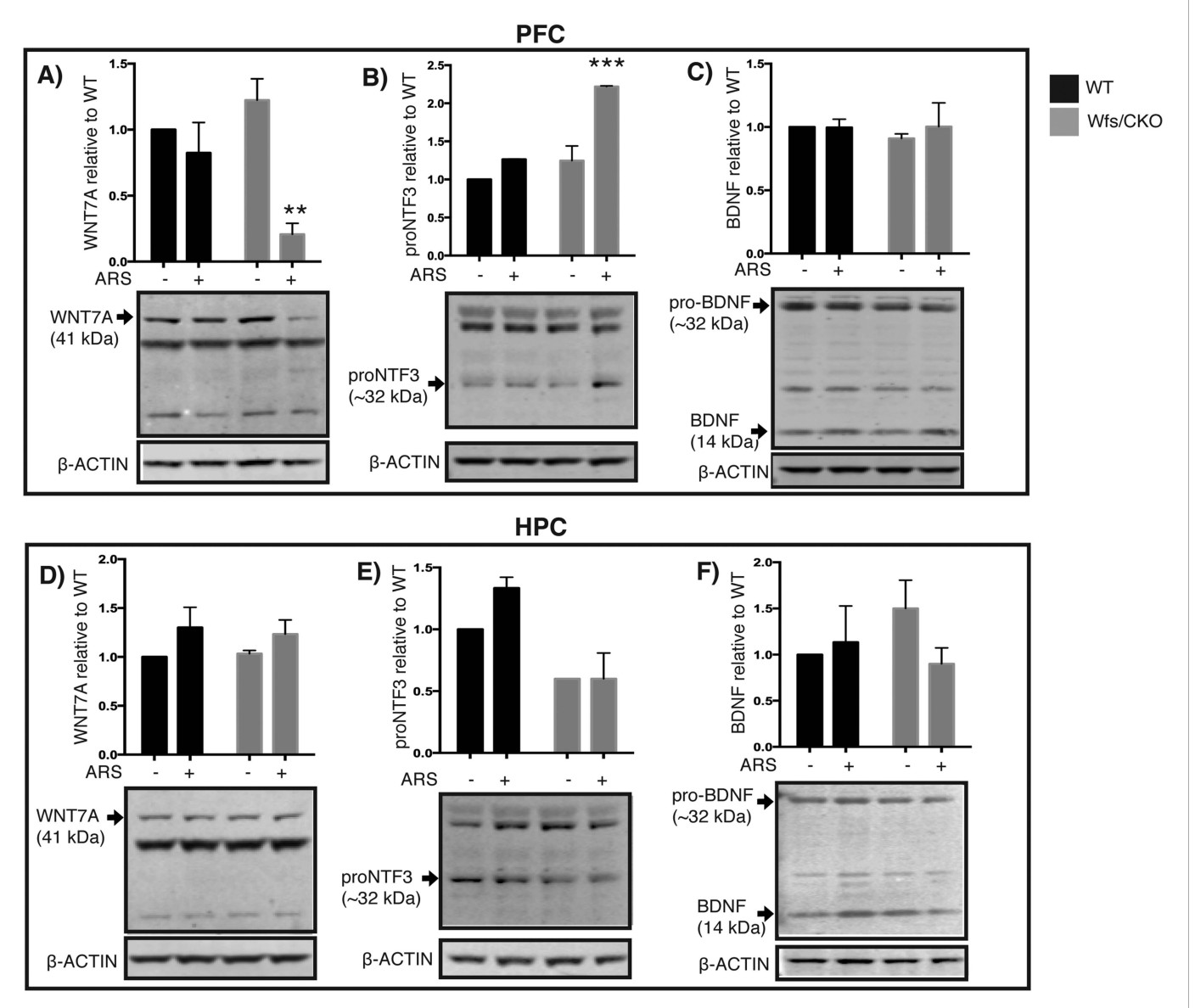

**Figure 7**. Wfs1 is required for normal endoplasmic reticulum function in the prefrontal cortex. Abundance of secreted polypeptides specifically enriched in the layer 2/3 pyramidal neurons is altered in the prefrontal cortex of Wfs1/CKO mice following ARS. (**A**) In the PFC lysates, WNT7a is significantly reduced in the Wfs1/CKO mice exposed to ARS (Two-way ANOVA—Bonferroni post hoc test: CKO Basal vs CKO + ARS, p < 0.001**, N = 3 per group). (**B**) proNTF3, the 23 kDa precursor of NTF3, is strongly elevated in Wfs1/CKO mice exposed to ARS indicating dysregulated precursor processing (Two-way ANOVA—Bonferroni post hoc test—CKO Basal vs CKO + ARS, p < 0.0001***, N = 3 per group). (**C**) BDNF, whose expression is not restricted to the layer 2/3 pyramidal cells in the prefrontal cortex, is not significantly altered. Abundance of the same polypeptides, WNT7a, proNTF3 and BDNF are not altered in the hippocampus (HPC) lysates regardless of the genotype and exposure to stress (**D**, **E**, **F**).

consistently observed in Wfs1 knockout animals in response to ARS (Two-way ANOVA—Bonferroni post hoc test: CKO Basal vs CKO + ARS, p < 0.0001***, N = 3 per group) (**Figure 7B**). These changes were not observed in hippocampal neurons extracts processed from the same animals (**Figure 7D,E**). No changes in BDNF were evident in Western blots prepared from either the PFC or hippocampus (HPC) in response to loss of *Wfs1* (**Figure 7C,F**). These data demonstrate that processing and release of growth factors, neurotrophins and perhaps other secreted proteins are disturbed in layer 2/3 neurons harboring *Wfs1* deletion. This is consistent with the localization of WFS1 to the ER of these cells (**Figure 2**) and with previous studies of WFS1 function in pancreatic β cells.

## Discussion

MDD is a behaviorally and genetically complex disorder that that is thought to arise from dysfunction in a distributed circuit involving the mPFC, the anterior cingulate cortex, the striatum, the thalamus, and other brain areas (*Russo et al., 2012*). Although the cell types and circuit elements that underlie different aspects of depressive behaviors and the actions of antidepressant therapies are beginning to be identified, the impact of environmental stress on depression circuitry has not been investigated thoroughly. Here, we report that *Wfs1*, a gene that can cause depression (*Flint and Kendler, 2014*), is co-expressed with *Ntf3*, *Penk*, and *Wnt7a* in supragranular 2/3 pyramidal neurons in the mPFC. We demonstrate that conditional deletion of Wfs1 from the forebrain in general, or from the mPFC specifically, results in the induction of depressive behaviors in response to acute stress. This is accompanied by hyperactivation of the HPA axis and elevation of serum corticosterone. We show that Wolframin (the protein encoded by *Wfs1*) is localized to the ER in these neurons and that its loss results in altered growth and trophic factor processing/secretion. Finally, we employ viral tracing methodology to determine that *Wfs1*-expressing cells in the mPFC project within the cortex and to the caudate putamen and that they receive afferent information from other cortical areas, from the amygdala and from the posterior nuclear thalamic group. We conclude that layer 2/3 pyramidal cells in the mPFC are a crucial node in the circuitry regulating depressive behaviors in response to stress and that identification of mechanisms that regulate the activity of these neurons, and their downstream targets can provide important insights into the relationship between stress and depression.

### *Wfs1* and the brain

Patients with Wolfram syndrome suffer from a broad spectrum of psychiatric and neurological problems, including suicidal tendencies. Heterozygous carriers of mutant *Wfs1* alleles are also affected, exhibiting clinical features of pure MDD. It has been estimated that as many as 7% of the patients hospitalized for MDD may be predisposed due to *Wfs1* mutations (*Swift et al., 1990*; *Crawford et al., 2002*; *Swift and Swift, 2005*). Studies of *Wfs1* knockout mice have shown that they replicate central and peripheral features of Wolfram syndrome and that they display impaired behavioral adaptation to stress (*Sequeira et al., 2003*; *Kato et al., 2008*; *Luuk et al., 2009*). Interpretation of these data with regard to brain circuitry is confounded by the complexity of the phenotypes present in *Wfs1* knockout animals and by the expression of *Wfs1* in specific subsets of neurons in the cerebral cortex, the hippocampus, the nucleus accumbens (NAc), the thalamus, the amygdala, and the brainstem. Our demonstration that *Wfs1* is required for normal ER function in layer 2/3 pyramidal cells in the mPFC and that these cells moderate depressive behaviors in response to stress, raises two issues regarding *Wfs1*. First, given previous studies documenting an essential function for Wfs1 in pancreatic β cells (*Ishihara et al., 2004*; *Fonseca et al., 2005*) and the retina (*Bonnet Wersinger et al., 2014*), and detailed biochemical studies demonstrating that WFS1 negatively regulates ER stress signaling through ATF6α (*Fonseca et al., 2010*), it seems probable that cell types expressing *Wfs1* in the brain also require its function to regulate ER stress. In the case of pancreatic β cells, it is thought that WFS1 tightly controls ER stress to regulate insulin production in response to frequent fluctuations of blood glucose. Given these examples, the expression of *Wfs1* in selected cell types may reflect a requirement for WFS1 in regulating ER function, perhaps even secretory function, in response to extracellular signals that modulate the activity of these cell types. In this context, the observations that WNT7A and proNTF3 protein levels are altered in the cortex of Wfs1/CKO animals in response to stress may be important. Clear positive roles for WNT7A in synapse formation and maintenance have been documented in several systems (*Hall et al., 2000*; *Gogolla et al., 2009*). In contrast to NTF3, which has an important trophic role in synapse formation, proNTF3 can induce apoptosis in cultured cells, suggesting a negative role for proNTF3 in neurons (*Yano et al., 2009*). While it is notable that both decreased WNT7A and increased proNTF3 could result in loss of synaptic trophic support in response to stress, this is only one of many possible pathways that could be negatively impacted in layer 2/3 neurons as a consequence of Wfs1 deletion.

If neurons expressing *Wfs1* in the brain require its function to respond to and moderate behavior in response to specific signals, one might ask which signals carry this information to Wfs1-positive neurons in the context of stress. We find that receptors for three circulating factors that are important for stress responsivity are expressed in *Wfs1*-positive cells in the cortex: *Nr3c1*, the glucocorticoid

receptor; *Esr1*, the estrogen receptor; and several adrenergic receptors, *Adra2c*, *Adr2a* and *Adrb1*. It will be of interest, therefore, to determine in future studies whether loss of *Wfs1* regulates the expression or function of these receptors in layer 2/3 pyramidal cells. This avenue of investigation could lead to an improved understanding of signals that are particularly relevant to the regulation of depressive behaviors by the mPFC.

### mPFC circuitry and the regulation of stress responses

We demonstrate here that conditional deletion of *Wfs1* from the forebrain in general, or from the mPFC specifically, results in enhanced sensitivity to ARS. The data demonstrate that the PVN is hyperactivated in response to stress in these animals, and that this leads to elevated peripheral corticosterone, is interesting with regard to the connectivity of *Wfs1*-expressing neurons in the mPFC. As illustrated in *Figure 6*, *Wfs1*-expressing cells in the mPFC receive input from several adjacent cortical regions, from the BLA and from the thalamus. Taken together with the expression of the glucocorticoid, estrogen, and adrenergic receptors discussed above, these pathways provide ample opportunity for modulation of Wfs1 cells in the context of stress. Thus far, our data have demonstrated projections from *Wfs1*-expressing cells primarily to neighboring and contralateral cortical structures, and to the caudate putamen. Given the central role that striatal circuits play in both stress and depression, the simplest interpretation of our results is that loss of *Wfs1* in mPFC impairs signaling to the striatum, and that failure to stimulate GABAergic output from the NAc to the PVN (*Russo et al., 2012*) is responsible for hyperactivation of the HPA axis in this model of stress-induced depression. Alternatively, projections from the mPFC to other subcortical sites (e.g., the amygdala) might be altered in response to Wfs1 loss in layer 2/3 neurons. It will be of interest, for example, to determine whether the activity of S100a10 corticostriatal neurons that express p11 is altered in these animals in altered in response to local circuit disruptions as a result of loss of Wfs1 (*Svenningsen et al. 2006*; *Schmidt et al., 2012*). Additional studies will be required to understand fully the properties of the mPFC circuits that are engaged by Wfs1-expressing pyramidal cells.

### Stress-induced depression

Two major clinically relevant findings stimulated us to conduct these studies: the identification of *Wfs1* as a gene that can be causative for MDD (*Flint and Kendler, 2014*); and the dynamic structural alterations to dendritic arbors of superficial layer pyramidal cells that have been documented in human postmortem and animal studies in response to stress (*Rajkowska et al., 1999*; *Radley et al., 2004*). The data we have presented here demonstrate layer 2/3 pyramidal neurons in the mPFC that have been compromised or sensitized by removal of *Wfs1* exacerbate depressive behaviors in response to ARS. One interpretation of these data that is consistent with known findings is that normal function of these cells is required to moderate stress and suppress depressive behaviors. Although genetic studies indicating an association between MDD and bipolar disorder and Wfs1 have not been consistently confirmed (*Ohtsuki et al., 2000*; *Koido et al., 2005*), investigation of Wfs1 carriers and family members has provided strong evidence that Wfs1 mutations can lead to an increased incidence of psychiatric admissions for hospitalization (*Swift et al., 1991*). Given the important contributions that family studies have made to our understanding of disease (*Pohodich and Zoghbi, 2015*), we believe that further investigation of layer 2/3 cells in other models of depression, and in response to a variety of stressors, can shed important new light on pathways that operate in these cells to moderate depressive behaviors in response to stressful environments.

## Materials and methods

### Animals

All animal protocols were carried out in accordance with the US National Institutes of Health Guide for the Care and Use of Laboratory Animals and were approved by the Rockefeller University Institutional Animal Care and Use Committee. All mice were raised at 78˚F in 12-hr light:12-hr dark conditions with food and water provided ad libitum except when noted, for experiments requiring food restriction. The Ntf3 bacTRAP mice and Wfs1-mutant mice were generated and maintained at The Rockefeller

University. The Ntf3 bacTRAP transgenic mice were generated by modifying a BAC clone containing the Ntf3 gene to insert an EGFP/L10a fusion protein into the translation start site, and the modified BAC was used for transgenesis as described previously (Gong et al., 2003). Conditional Wfs1 KO (Wfs1/CKO) mice were generated by breeding floxed Wfs1 mice with Emx1.IRES.Cre mice—(Stock 005628) purchased from the Jackson Laboratory (Bar Harbor, ME). Wfs1:Cre ER$^{T2}$ Tg2 mice (Stock 009614) were also obtained from Jackson Laboratory and fed 400 mg/kg tamoxifen diet (Teklad lab diets, Harlan Pharmaceuticals, Frederick, MD) from 2 weeks before surgery onwards to activate Cre recombinase.

## Molecular cloning, BAC modification and transgenesis

The BAC modification and transgenesis were carried out as previously established (Gong et al., 2003). Using CloneFinder, BAC RP23-79E22 incorporating 170 kB of genomic DNA at the minus strand of Chr 6: 125,999,011–126,169,212 and containing Ntf3 gene and regulatory elements was selected for modification. A 711 bp homology arm ('A' box) immediately upstream of Ntf3 translational start codon was PCR amplified using the forward primer (GGCGCGCCTAGACAGCTT GAACTAACTGTG) and reverse primer (CTGCTGGGTAAGGAGAGGAGCCTT) and cloned into the pS296 shuttle vector containing the EGFP/L10a transgene using the AscI and NotI restriction sites. The S296 vector was electroporated into the DH10B bacteria containing the RP23-79E22 BAC and pSV1.RecA plasmid and allowed to grow overnight before recombination was terminated by growth at 43°C. BAC co-integrates were screened for successful recombination by PCR and Southern blot analysis of Hind III digested BAC DNA using radiolabeled 'A' box as a probe. The modified BAC was then grown and prepared by double acetate purification with CsCl gradient centrifugation, followed by membrane dialysis. The quality and concentration of purified BAC DNA was determined by pulsed field gel electrophoresis. Modified BAC DNA was injected into the pronuclei of fertilized FVB/N mouse oocytes at a concentration of 0.5 ng/μl. Five transgenic founder lines were generated. Founders were crossed to C57Bl/6J wt mice and F1 progeny were screened for proper transgene expression by EGFP immunohistochemistry.

## Generation of rabbit polyclonal antibody to WFS1

Rabbit polyclonal antibody was generated against a custom synthesized peptide (CEPPRAPRPQAD-PSAG) of WFS1 (Green Mountain Antibodies, Burlington, VT). The antigen was purified to 90% by HPLC and coupled to a carrier before being injected to rabbits. The antibody titer from the rabbit blood was measured and effective concentration of the antibody for immunohistochemistry was determined empirically.

## Immunohistochemistry

Brains were processed identically with MultiBrain Technology (NSA, NeuroScience Associates, Knoxville, TN) for DAB immunohistochemistry with a 1:75,000 dilution of Goat anti-EGFP serum (Heiman et al., 2008) according to the Vectastain elite protocol (Vector Labs, Burlingame, CA). Serial sections were digitized with a Zeiss Axiosko2 microscope at 10× magnification. For immunofluorescent staining, mice were deeply anesthetized using $CO_2$ chamber and transcardially perfused with 10 ml of phosphate-buffered saline (PBS) followed by 30 ml of 4% paraformaldehyde (PFA) in PBS. Brains were post-fixed in 4% PFA for 1 hr and cryoprotected by sequential sinking in 5% wt/vol sucrose in PBS at 4°C for 24 hr with gentle agitation followed by 30% wt/vol sucrose in PBS for the next 24 hr. 40-μm coronal sections were cut with Leica SM200R freezing microtome. Sections were blocked in 5% Normal Donkey Serum in PBS/0.1% Triton X-100 for 30 min and incubated overnight at 4°C with primary antibody against chicken anti-EGFP (Abcam, Cambridge, MA; 1:500), rabbit anti-WFS1 (Green Mountain Antibodies; 1:1000), mouse anti-NeuN (Millipore, Billerica, MA; 1:500), or anti-cFOS (Santa Cruz, Dallas, TX; 1:1000) in the blocking buffer. Appropriate Alexa dye-conjugated secondary antibodies were used at 1:400 dilution in the blocking buffer. Sections were mounted on the SuperFrost slides (VWR, South Plainfield, NJ) using Fluorogel media containing DAPI for nuclear counterstain (EMS, Hatfield, PA). All sections were imaged on Zeiss LSM-700 confocal microscope.

## Generation of Wfs1 conditional null allele

The Wfs1-mutant mice were generated at Lexicon Pharmaceuticals, Inc (The Woodlands, TX). The conditional targeting vector was derived using the Lambda KOS system. The Lambda KOS phage

library, arrayed into 96 superpools, was screened by PCR using exon 8-specific primers Wfs2 [5′-GTGAAGTACCCTTTACACGC-3′] and Wfs3 [5′-GCAGCAGGTCGGTGAGAG-3′]. The PCR-positive phage superpools were plated and screened by filter hybridization using the 280-bp amplicon derived from primers Wfs2 and Wfs3 as a probe. Three pKOS genomic clones, pKOS-12, pKOS-39, and pKOS-58 were isolated from the library screen and confirmed by sequence and restriction analysis. Gene-specific arms (5′-GAGGCCCAGGAGTGGGAAAGTCTAGGGTGTG-3′) and (5′-GACAAGGCTCCCTG TAATCAAACCAGAAGG-3′) were appended by PCR to a yeast selection cassette containing the URA3 marker. The yeast selection cassette and pKOS-12 were co-transformed into yeast, and clones that had undergone homologous recombination to replace a 2706 bp region containing exon 8 with the yeast selection cassette were isolated. This 2706 bp fragment was independently amplified by PCR and cloned into the intermediate vector pLFNeo introducing flanking loxP sites and a Neo selection cassette (Wfs1–pLFNeo). The yeast cassette was subsequently replaced with the Wfs1–pLFNeo selection cassette to complete the conditional Wfs1-targeting vector that has exon 8 flanked by loxP sites. The NotI-linearized targeting vector was electroporated into 129/SvEv[Brd] (Lex-1) ES cells. G418/FIAU-resistant ES cell clones were isolated, and correctly targeted clones were identified and confirmed by Southern analysis using a 485-bp 5′ external probe (9/50), generated by PCR using primers Wfs9 [5′-CTGCCTTGCTTGCAATGTTG-3′] and Wfs50 [5′-CATGTCCAAGACAGGATGTG-3′], and a 443-bp 3′ external probe (37/53), amplified by PCR using primers Wfs37 [5′-CAACATTT CTCAGAGCTTCC-3′] and Wfs53 [5′-CGTGTTAGAGTGCTGTACAG-3′]. Southern analysis using probe 9/50 detected a 12.1-kB wt band and 9.6-kB mutant band in Hind III digested genomic DNA while probe 37/53 detected an 8.8-kB wt band and 10.8-kB mutant band in Bam HI digested genomic DNA. Three targeted embryonic stem (ES) cell clones were microinjected into C57BL/6 (albino) blastocysts. The resulting chimeras were mated to C57BL/6 (albino) females to generate mice that were heterozygous for the *Wfs1* conditional mutation (*Wfs1* F/+).

## Stereotaxic intracranial injections

All animals were anesthetized by intraperitoneal injection of a cocktail of ketamine (100 mg/ml) and xylazine (1 mg/ml), at a volume equivalent to 10% of their body weight. The stereotaxic coordinates for mPFC were determined using Paxinos/Franklin mouse atlas (*Paxinos and Franklin, 2001*). The following stereotaxic measurements were taken relative to the bregma and with the depth determined from the brain surface: anterior–posterior: +2.46, ML: ±0.75, DV: 1.75. For local Cre-mediated deletion of *Wfs1* in the PFC, 0.5 μl of AAV2.CMV.HI.GFP-Cre.SV40 or control reporter virus AAV2.CMV.PI.EGFP.WPRE.bGH was injected bilaterally into the medial PFC. 15 days after injection of AAV vectors, mice were subjected to behavior assays or euthanized, and their tissues harvested for expression analysis.

For anterograde tracing, Wfs1::Cre ER[T2] Tg2 mice were injected with 0.5 μl of Cre-dependent AAV viral tracer, AAV2.EF1a.DIO.eYFP.WPRE.hGH, in mPFC to trace the projections of *Wfs1*-expressing cells. For monosynaptic retrograde tracing, a helper virus expressing Cre-dependent eGFP and an avian receptor protein, TVA (AAV9-pEF1a-FLEX-GT), was injected into mPFC of Wfs1:CreER[T2] mice, and allowed to express for 6 weeks, followed by injection of glycoprotein deleted rabies virus expressing mCherry [(EnvA)SAD-dG-mCherry] that allowed specific retrograde labeling of neurons making direct monosynaptic connections to the Wfs1 cells in mPFC.

## Affinity purification of translating ribosomes

All polysome purifications and mRNA extractions were performed as described previously (*Heiman et al., 2008*). Three adult mice, balanced for gender and 8–10 weeks old, were pooled for each sample, and three biological replicates were collected for each bacTRAP line. Briefly, cortices were rapidly dissected in ice-cold HBSS containing 2.5 mM HEPES-KOH (pH 7.4), 35 mM glucose, 4 mM NaHCO$_3$, and 100 μg/ml cycloheximide, homogenized in extraction buffer 10 mM HEPES-KOH (pH 7.4), 150 mM KCl, 5 mM MgCl$_2$, 0.5 mM DTT, 100 μg/ml cycloheximide, RNAs in RNAse inhibitors (Promega, Madison, WI) and complete-EDTA-free protease inhibitors (Roche, Madison, WI), and centrifuged at 2000×*g* for 10 min. The supernatant from the homogenate was supplemented with a mix of detergents DHPC (Avanti Polar Lipids, Alabaster, AL) and NP-40 (Sigma–Aldrich, St Louis, MO), and cleared by a second centrifugation at 20,000×*g* for 15 min. Polysomes were immunoprecipitated with 100 μg of monoclonal anti-EGFP 19C8 and 19F7 antibodies bound to

Protein-L conjugated Streptavidin MyOne Dynabeads overnight at 4°C and washed with high salt buffer containing 10 mM HEPES-KOH (pH 7.4), 350 mM KCl, 5 mM MgCl$_2$, 1% NP-40, 0.5 mM DTT, 100 µg/ml cycloheximide, and RNAs in RNAse inhibitors (Life Technologies, Grand Island, NY). Bound RNA was extracted and purified using Absolutely NanoPrep RNA purification kit (Stratagene, La Jolla, CA). RNA quantity was measured using Nanodrop 1000 spectrophotometer (Wilmington, DE) and quality determined with Agilent 2100 Bioanalyzer. RNA from 50 µl of homogenate prior to immunoprecipitation was also purified as Cortex input samples.

## Specificity index analysis of translational profiles

15 ng of total RNA was amplified with GeneChip Expression 3′ Amplification 2-cycle cDNA Synthesis kit (Affymetrix, Santa Clara, CA) and hybridized to GeneChip Mouse Genome 430 2.0 microarrays (Affymetrix) according to the manufacturer's instructions. CEL files for all RNA samples were imported into GeneSpring GX 12.6 (Agilent, Santa Clara, CA) and preprocessed with the GCRMA algorithm. For each experiment, three biological replicates of IP were averaged and compared with the Cortex Input using an X–Y scatter plot for all 40,101 probe sets. Thresholds were set at 40th percentile for expression and for Top 150 probe sets that had the highest fold change difference in the target cell type compared with the Whole Cortex input.

For specificity index analysis of multiple distinct cell types, microarray data were preprocessed using Bioconductor package affy (*Gautier et al., 2004*) and normalized using Robust Multichip Average (*Irizarry et al., 2003*). Differentially expressed genes among the different cell types were identified by using Bioconductor package limma (*Smyth, 2005*). p-values were corrected to control the false discovery rate of multiple testing according to the Benjamini–Hochberg procedure (*Benjamini and Hochberg, 1995*) at 0.05 threshold. Significant genes with fold change (|fold change| ≥ 1) were considered top genes. Cell-specific gene expression analysis was performed by using the specificity index statistic (pSI) R-package version 1.1 (*Dougherty et al., 2010*). The pSI allows identification of groups of genes enriched in certain cell type compared with other cell types.

## Quantitative real-time RT-PCR

cDNA was synthesized from 15 ng of total RNA from three IP replicates and Whole cortex input using WT-Ovation RNA Amplification kit (NuGEN Technologies, San Carlos, CA) and then purified with the Qiagen Quick PCR cleanup, according to the manufacturer's instructions (Qiagen, Valencia, CA). PCR was carried out in Applied Biosystems (Carlsbad, CA) StepOnePlus RT-PCR System using Taqman probes for *Ntf3* (Mm00435413_s1), *Wfs1* (Mm00495979_m1), *Penk* (Mm01212875_m1), and *Gapdh* (Mm99999915_g1) (Applied Biosystems) and PerfeCTa qPCR Fastmix II ROX (Quanta Biosciences, Gaithersburg, MD) following standard cycling conditions (50°C for 2 min, 95°C for 10 min, 40 cycles of 95°C for 15 s and 60°C for 1 min). Products that did not yield a product in at least 2 replicates prior to 35 cycles were excluded from further analysis. All data were normalized to *Gapdh*, and relative expression changes between conditions were calculated with the ddCT method (*Livak and Schmittgen, 2001*).

## Western blot

Brain tissue pooled from 2 animals was quickly dissected in ice-cold PBS. Three biological replicates were collected for each condition. The prefrontal cortex and hippocampus were separately homogenized in 500 µl of lysis buffer containing 50 mM Tris HCl (pH 8.0), 20 mM EDTA, 2500 units of Benzonase nuclease (Sigma–Aldrich) and complete-EDTA-free protease inhibitors (Roche). The lysate was cleared by centrifugation at 2000×g and supplemented with triple detergents including 10% sodium deoxycholate, 10% NP-40, and 1% SDS by volume. Samples were centrifuged a second time at maximum speed (20,000×g) for 15 min. The protein concentration of samples was determined using the QuBit 2.0 protein quantitation kit (Invitrogen, Grand Island, NY).

30 µg of protein was denatured and loaded in each well of 4–12% NuPAGE Bis-Tris reducing gels (Life Technologies). The gels were run at 100 V and subjected to wet transfer. PVDF membranes were first activated with methanol and equilibrated in transfer buffer containing 1× transfer buffer (Life Technologies) and 20% methanol. The bands were transferred overnight from gels to PVDF membranes using chilled transfer buffer at 4°C. Membranes were blocked in Odyssey blocking buffer (LI-COR, Lincoln, NE) for 30 min to prevent non-specific background binding. Following primary

antibodies were diluted in OBT buffer containing Odyssey blocking buffer and 0.1% Tween-20: rabbit anti-NTF3 (Abcam; 1:100), rabbit anti-BDNF (Santa Cruz; 1:100), rabbit anti-PENK (Abcam; 1:100), and rabbit anti-β ACTIN (Abgent, San Diego, CA; 1:2000).

Membranes were blotted with primary antibody overnight at 4°C and washed three times with PBST buffer containing 1× PBS and 0.1% Tween-20 for 10 min each wash. IR-dye-conjugated secondary antibodies (LI-COR; 1:40,000) were diluted in OBT buffer and blotted on to membranes for 1 hr at room temperature followed by washes with the PBST buffer. Membranes were scanned using the Odyssey CLx Infrared Imaging System (LI-COR) and bands were quantified using the Image Studio Lite software (LI-COR).

## Behavioral analysis

Locomotor and exploratory behaviors were recorded for each individual mouse for 60 min using eight Digiscan open field (OF) apparatus and Fusion software (Accuscan Instruments, Inc., Columbus, OH). Mice were habituated to the testing room in their homecages for 30 min before the start of the experiment. A large arena (50 cm × 50 cm × 22.5 cm) equipped with infrared photocells at two different levels (20 and 50 mm above the floor) was used to record horizontal locomotor activity (total distance), and thigmotaxis (time spent in the center vs total distance). Two fluorescent lamps positioned on two sides of the room provided light levels of about 450 lux in the OF arenas. Each animal was placed in the center of the OF, and its activity was measured for 60 min.

SPT was performed with a two-bottle procedure, during which mice had free access to both water and sucrose solution. The fluids were provided in 50-ml falcon tubes containing stoppers fitted with ball–point sipper tubes to prevent leakage. Group-housed animals were first habituated for 3 consecutive days to 1% sucrose solution. After habituation, mice were individually housed and water deprived for 20 hr. On the test day, mice were presented with two-bottle choice conditions with either 2% sucrose (in drinking water) or drinking water alone. Consumption of water or sucrose solution was measured by weighing the bottles before and after the test. Bottles were counterbalanced across the left and the right sides of the cage. Sucrose preference was determined as the ratio of average sucrose solution intake (in ml) to the average total fluid intake (in ml).

For Porsolt's FST, mice were habituated to the testing room in their homecages for 30 min and then individually placed in a glass cylinder (16 cm diameter, 25 cm height) filled with tap water (23–25°C) to a height of 20 cm. Test sessions lasted 6 min and were videotaped. The episodes of immobility were scored every 5 s for the final 4 min of the test.

Elevated plus maze consisted of two opposite open arms without sidewalls and two enclosed arms of the same size with 14-cm high sidewalls and an endwall. The arms extended from a common central square (5 cm$^2$ × 5 cm$^2$) perpendicular to each other, making the shape of a plus sign. The entire plus-maze apparatus was elevated to a height of 30 cm. Testing began by placing an animal on the central platform of the maze facing the closed arm. An arm entry was counted only all the four limbs were within a given arm. Standard 10-min test duration was applied and the maze was wiped with 70% ethanol in between trials. Test sessions were video recorded with a camera in the center of the maze while ensuring minimal shadows. EthoVision software was used to record the time spent on open arms and closed arms, total distance moved, and number of open arm and closed arm entries.

For appetite operant conditioning, mice were food restricted to 85–90% of their baseline body weight. Mice were introduced with dust-free precision pellets (Bio-Serv, Flemington, NJ) in their homecages to eliminate novelty associated with these pellets. Operant conditioning were carried out in 12 identical chambers (Med Associates, St Albans City, VT) equipped with a food magazine, house light, stimulus light, tone generator, and three nose-poke apertures. Mice were given 2 days of magazine training with variable interval 20 for time of pellet dispensal until a total of 50 reinforcers were delivered. For acquisition of instrumental learning, one of the three illuminated nose-poke apertures was designated as active (left, center, or right) in a pseudo-randomized and balanced manner for each genotype. A response in the active aperture (correct response) resulted in delivery of a food pellet, accompanied by a 3-s presentation of stimulus light above the magazine and tone. A response in the other two apertures (incorrect response) had no effect. The food reinforcers were all delivered on a variable ratio-2 schedule (1, 2 or 3 correct responses were

required to obtain reinforcement). Mice were tested for 25 min each day over 12 consecutive days of training (Acquisition), and the number of correct and incorrect responses were recorded. On the 13th day, the far right or far left aperture that was previously unreinforced was now designated to be the active aperture. Reversal learning and thus behavioral flexibility was measured over the course of next 4 consecutive days.

Morris water maze testing took place in a white circular Persplex pool, 97 cm in diameter, surrounded by proximal and distal extramaze cues. The pool was filled with opaque water maintained at 23.5°C. Mice were shaped in the tank 1 day prior to testing using a four trial procedure in which a smaller ring (55 cm) was placed inside of the larger (97 cm) ring to decrease the total swimming area. Mice were first placed on a visible 10 × 10 cm platform for 10 s and then removed. They were then placed at three distances progressively further from the platform and allowed to swim to the platform. No data were collected during shaping. From the second day onwards, a transparent lucite platform was submerged just underneath the surface of water and placed in the same location for all trials. 6 trials were conducted per day for 4 consecutive days. Each mouse was placed in one of the four start positions, which varied for each trial, and given 120 s to find the platform in each trial. If the mouse did not find the platform within this time, then it was led to the platform where it was allowed to stay for 10 s. Each mouse was returned to its homecage during the inter-trial interval of approximately 20 min. On the seventh day, task retention was assessed by removing the platform from the pool. Throughout the trials, swim time, that is, latency to reach platform (in s), swim distance (in cm), and swim speed (in cm/s) were recorded. All data for Morris water maze were collected using EthoVision XT (Noldus, Leesburg, VA).

## Corticosterone RIA

Submandibular blood of five wt and five Wfs1/CKO mice were collected in EDTA-coated tubes (Fisher Scientific, Pittsburgh, PA), centrifuged at 12,000×$g$ for 5 min, and plasma was extracted. RIA was carried out on the plasma following the manufacturer's protocols (MP Biomedicals, Solon, OH). Residual I-125 activity was measured using a Gamma counter.

## Quantification of cFOS-labeled cells

Light microscopy and simple cell counting methods were used to measure cFos immunoreactivity in brain sections. The number of dots representing cFos protein in immunostained brain sections was counted for every 40 μm through the whole area of PVN using the Cell Count plugin of the ImageJ software. Immunopositive cFos cells from brain sections of three animals were counted and averaged, and standard error of mean was calculated.

## Statistics

Prism 6 software (GraphPad, La Jolla, CA) was used for statistical analysis of behavioral, histological, and biochemical data. Values are represented as mean ± standard error of mean. The cutoff set for significance for all experiments was α <0.05. For data involving two or more independent variables, two-way ANOVA was used, and Bonferroni post hoc tests, correcting for multiple comparisons, were used. Repeated measures one-way ANOVA was used for behavior assays involving multiday training of same mice such as in Morris water maze and appetite operant conditioning paradigms. Unpaired Student's $t$-test was used for quantitative RT-PCR data. pSI R-package version 1.1 was used for specificity index analysis of translational profiles of multiple cell types.

## Acknowledgements

This work was supported by the Howard Hughes Medical Institute, NIH/NIMH 5P50MH090063 P2 and NIH/NIDA 5P30DA035756. We thank Judy Walsh for administrative support, Jie Xing and Jenna Rimberg for their skilled assistance, Eric Schmidt, Jodi Gresack, Erika Andrade, and all other Heintz lab members for their helpful suggestions. We would like to thank Wenxiang Zhang from the Rockefeller University Genomics Resource Center for help with the TRAP data production and Kunihiro Uryu from the Electron Microscopy Resource Center for help with generating electron micrographs. We would also like to thank Christopher Wilson for help with corticosterone RIA assays and Bruce McEwen for insightful advice on stress paradigms.

# Additional information

## Funding

| Funder | Grant reference | Author |
|---|---|---|
| Howard Hughes Medical Institute (HHMI) | | Nathaniel Heintz |
| National Institutes of Health (NIH) | NIH/NIMH 5P50 MH090063 P2 | Nathaniel Heintz |

The funders had no role in study design, data collection and interpretation, or the decision to submit the work for publication.

## Author contributions

PS, Conception and design, Acquisition of data, Analysis and interpretation of data, Drafting or revising the article; AM, Analysis and interpretation of data, Drafting or revising the article; NH, Conception and design, Analysis and interpretation of data, Drafting or revising the article

## Ethics

Animal experimentation: All animal protocols were carried out in accordance with the US National Institutes of Health Guide for the Care and Use of Laboratory Animals and were approved by the Rockefeller University Institutional Animal Care and Use Committee.

# Additional files

## Major datasets

The following dataset was generated:

| Author(s) | Year | Dataset title | Dataset ID and/or URL | Database, license, and accessibility information |
|---|---|---|---|---|
| Shrestha P, Mousa A, Heintz N | 2015 | Expression data from cerebral cortices of bacTRAP transgenic mice | http://www.ncbi.nlm.nih.gov/geo/query/acc.cgi?acc=GSE69340 | Publicly available at the NCBI Gene Expression Omnibus (Accession no. GSE69340). |

The following previously published datasets were used:

| Author(s) | Year | Dataset title | Dataset ID and/or URL | Database, license, and accessibility information |
|---|---|---|---|---|
| Nakajima M, Görlich A, Heintz N | 2014 | Comparative analysis of different cortical interneuron groups | http://www.ncbi.nlm.nih.gov/geo/query/acc.cgi?acc=GSE56996 | Publicly available at the NCBI Gene Expression Omnibus (Accession no. GSE56996). |
| Schmidt EF, Warner-Schmidt JL, Otopolik BG, Pickett SB, Greengard P, Heintz N | 2012 | Comparative analysis of S100a10 and Glt25d2 cortical pyramidal cells | http://www.ncbi.nlm.nih.gov/geo/query/acc.cgi?acc=GSE35758 | Publicly available at the NCBI Gene Expression Omnibus (Accession no. GSE35758). |
| Doyle JP, Dougherty JD, Heiman M, Schmidt EF, Stevens TR, Ma G, Bupp S, Shrestha P, Shah RD, Doughty ML, Gong S, Greengard P, Heintz N | 2008 | Application of a translational profiling approach for the comparative analysis of CNS cell types | http://www.ncbi.nlm.nih.gov/geo/query/acc.cgi?acc=GSE13379 | Publicly available at the NCBI Gene Expression Omnibus (Accession no. GSE13379). |

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
