## [Decision Letter]

Thank you for submitting your work entitled “Layer 2/3 pyramidal cells in the medial prefrontal cortex moderate stress-induced depressive behaviors” for peer review at *eLife*. Your submission has been favorably evaluated by a Senior Editor, a Reviewing Editor, and two reviewers.

The reviewers have discussed their reviews with one another, and the Reviewing Editor has drafted this decision to help you prepare a revised submission.

This study from the Heintz lab presents a technically sophisticated analysis of *Wfs1* gene function in the superficial cortical layer 2/3 of the medial prefrontal cortex, and demonstrates a role for *Wfs1* in regulating stress responses and depression-related behaviors. Previous studies linked *Wfs1* with an increased risk for mood disorders including depression, but the cellular and molecular bases through which this gene may drive depression-related behaviors were not clear. To address these questions, the authors first profiled gene expression in layer 2/3 pyramidal cells of the mPFC using a novel bacTRAP mouse line and establish the selective expression of *Wfs1* in these neurons. Using novel conditional *Wfs1* CKO mice, they demonstrate that loss of *Wfs1* in mPFC cortical pyramidal neurons leads to depression related behaviors (immobility in the forced swim test and anhedonia) after acute restraint stress (ARS). The authors show that the CKO mice have normal baseline motor and learning behaviors and hence are selectively impaired in stress responses. The connectivity of mPFC *Wfs1*-expressing neurons was analyzed using genetically based tracing methods. They provide evidence that the CKO mice exhibit an exaggerated physiological response to ARS, consistent with hyperactivation of the HPA axis (e.g. elevated c-fos+ in hypothalamic neurons and increased serum corticosterone levels). Finally, they show that *Wfs1* CKO mice exhibit abnormal levels of two secreted proteins (WNT7A and NTF3) only upon exposure to ARS, suggesting stress-induced ER deficits. Overall, the study provides convincing evidence to demonstrate that *Wfs1* and *Wfs1*-expressing mPFC pyramidal neurons are regulators of stress responses in mice, and lends important support for a link between *Wfs1* and depression-related behaviors in model organisms and patients.

This study is among the first to dissect neural circuitry related to depression-related behaviors using genes that are genetically linked to the disease. The reviewers felt that this study is an important addition to the understanding of depression-related neurocircuits, and provides important insights into where *Wfs1* deficiency could elicit depression-related behaviors. The newly developed *Wfs1* conditional KO mice will also provide an invaluable tool for future studies of other *Wfs1*-expressing neurons in regulating depression-related behaviors.

With that said, there are several places where the authors overstate what is actually shown. They do not rule out that *Wfs1* deficiency in structures other than layer 2/3 also play a role in these behaviors; the amygdala being one particular target. Also, the layer 2/3 *Wfs1*-expressing cells surely also project within the columns of the mPFC; thus, their effect on stress responsiveness might result from disruption of processing within the mPFC, and hence other outputs from mPFC, including from non-Wfs1 cells. The authors should revise their discussion of their results to make these other possibly more indirect effects of their mutation clear. Similarly, although WFS patients showed increased risk for depression, there is no evidence from GWAS studies to show *Wfs1* variants have a significant association with MDD. The authors should modify their Abstract and Discussion to avoid stating that their findings have implications for MDD (e.g. in the Abstract: “ Our data identify superficial layer 2/3 pyramidal cells as critical for moderation of stress in the context of MDD…”).

---

## [Author Response]

*[…] There are several places where the authors overstate what is actually shown. They do not rule out that* Wfs1 *deficiency in structures other than layer 2/3 also play a role in these behaviors; the amygdala being one particular target*.

We have revised the text to provide further clarity regarding the specificity of the genetic manipulations that allow us to conclude that it is *Wfs1* dysfunction specifically in the mPFC that alters stress-induced depressive behaviors. We have explicitly stated (in the subsection “Selective deletion of *Wfs1* in cortical excitatory neurons”), for example, that recombination in *Emx1*Cre mice is restricted to the forebrain, and that *Wfs1* expression in subcortical structures such as the amygdala are unaffected (we have added arrows to Figure 3 and modified the legend to highlight this point). We have also highlighted the reference that demonstrated that *Wfs1* is expressed in layer 2/3 throughout the cortex, and further discussed the point that it is the injections of AAVs expressing Cre recombinase in mPFC and subsequent behavioral studies that allow us to conclude that the mPFC is critical for these behaviors (in the subsection “Deletion of *Wfs1* in the medial prefrontal cortex is sufficient to cause stress-induced depression related behavior”).

*Also, the layer 2/3* Wfs1*-expressing cells surely also project within the columns of the mPFC; thus, their effect on stress responsiveness might result from disruption of processing within the mPFC, and hence other outputs from mPFC, including from non-*Wfs1 *cells. The authors should revise their discussion of their results to make these other possibly more indirect effects of their mutation clear*.

We have provided further discussion of our results to indicate that. While we have identified one cell type in the mPFC that is important for moderation of depressive behaviors in response to stress, there are certainly other cell types within the brain that may be able to generate similar behavioral phenotypes. These would include non-Wfs1-expressing cells in the mPFC and elsewhere whose function is impacted by Wfs1-expressing cells (see the subsection “Stress-induced depression”), or cells that participate in parallel circuits that impact these behaviors. Nevertheless, we believe identification of this first cell type as an example of a circuit element that is relevant to both stress and depression is an important advance.

*Similarly, although WFS patients showed increased risk for depression, there is no evidence from GWAS studies to show* Wfs1 *variants have a significant association with MDD. The authors should modify their Abstract and Discussion to avoid stating that their findings have implications for MDD (e.g. in the Abstract: “ Our data identify superficial layer 2/3 pyramidal cells as critical for moderation of stress in the context of MDD…”)*.

We have provided further discussion of the genetics of MDD, and the role that studies of familial disorders have played in the elucidation of mechanisms and circuits that are relevant to more general, sporadic diseases (see Discussion). In the case of MDD, GWAS studies have been disappointing thus far, so we think that it is appropriate to mention the precedents for this type of approach and the value it has brought to the field. We have ‘softened’ the tone of the Abstract as requested in spite of the strong arguments that a significant fraction of *Wfs1* carriers exhibit a pure form of major depression.